# Unifying Gradient Estimators for Meta-Reinforcement Learning via Off-Policy Evaluation

**Yunhao Tang\***
Columbia University
yt2541@columbia.edu

**Tadashi Kozuno\***
University of Alberta
tadashi.kozuno@gmail.com

**Mark Rowland**
DeepMind London
markrowland@deepmind.com

**Rémi Munos**
DeepMind Paris
munos@deepmind.com

**Michal Valko**
DeepMind Paris
valkom@deepmind.com

## Abstract

Model-agnostic meta-reinforcement learning requires estimating the Hessian matrix of value functions. This is challenging from an implementation perspective, as repeatedly differentiating policy gradient estimates may lead to biased Hessian estimates. In this work, we provide a unifying framework for estimating higher-order derivatives of value functions, based on off-policy evaluation. Our framework interprets a number of prior approaches as special cases and elucidates the bias and variance trade-off of Hessian estimates. This framework also opens the door to a new family of estimates, which can be easily implemented with auto-differentiation libraries, and lead to performance gains in practice. We open source the code to reproduce our results[1].

## 1 Introduction

Recent years have witnessed the success of reinforcement learning (RL) in challenging domains such as game playing [1], board games [2], and robotics control [3]. However, despite such breakthroughs, state-of-the-art RL algorithms are still plagued by sample inefficiency, as training agents requires orders of magnitude more samples than humans would experience to reach similar levels of performance [1, 2]. One hypothesis on the source of such inefficiencies is that standard RL algorithms are not good at leveraging prior knowledge, which implies that whenever presented with a new task, the algorithms must learn from scratch. On the contrary, humans are much better at transferring prior skills to new scenarios, an innate ability arguably obtained through evolution over thousand of years.

Meta-reinforcement learning (meta-RL) formalizes the learning and transfer of prior knowledge in RL [4]. The high-level idea is to have an RL agent that interacts with a distribution of environments at meta-training time. The objective is that at meta-testing time, when the agent interacts with previously unseen environments, it can learn much faster than meta-training time. Here, faster learning is measured by the number of new samples needed to achieve a good level of performance. If an agent can achieve good performance at meta-testing time, it embodies the ability to transfer knowledge from prior experiences at meta-training time. meta-RL algorithms can be constructed in many ways, such as those based on recurrent memory [5, 6], gradient-based adaptations [4], learning loss functions [7–9], probabilistic inference of context variables [10–12], online adaptation of hyper-parameters within a single lifetime [13, 14] and so on. Some of these formulations have mathematical connections; see e.g. [15] for an in-depth discussion.

We focus on gradient-based adaptations [4], where the agent carries out policy gradient updates [16] at both meta-training and meta-testing time. Conceptually, since meta-RL seeks to optimize the

---

[1] https://github.com/robintyh1/neurips2021-meta-gradient-offpolicy-evaluation

35th Conference on Neural Information Processing Systems (NeurIPS 2021).

Table 1: Interpretation of prior work on higher-order derivative estimations as special instances of differentiating off-policy evaluation estimates. Given any off-policy evaluation estimate $\widehat{V}^{\pi_\theta}$ from the second row, we recover higher-order derivative estimates in prior work in the first row, by differentiating through the estimate $\nabla_\theta^K \widehat{V}^{\pi_\theta}$.

| Off-policy evaluation estimates | STEP-WISE IS [26] | DOUBLY-ROBUST [27] | TAYPO-1 [28, 29] | TAYPO-2 [29] |
|---|---|---|---|---|
| **Prior work** | **DiCE [21]** | **Loaded DiCE [22, 23]** | **LVC [24]** | **Second-order (this work)** |

way in which the agent adapts itself in face of new environments, it needs to *differentiate* through the policy gradient update itself. This effectively reduces the meta-RL problem into estimations of Hessian matrices of value functions.

**Challenges for computing Hessian matrix of value functions.** To calculate Hessian matrices (or unbiased estimates thereof) for supervised learning objectives, it suffices to differentiate through gradient estimates, which can be easily implemented with auto-differentiation packages [17–19]. However, it is not the case for value functions in RL. Intuitively, this is because value functions are defined via expectations with respect to distributions that themselves depend on the policy parameters of interest, whereas in supervised learning the expectations are defined with respect to a fixed data distribution. As a result, implementations that do not take this into account may lead to estimates [4] whose bias is not properly characterized and might have a negative impact on downstream applications [20].

Motivated by this observation, a number of prior works suggest implementation alternatives that lead to unbiased Hessian estimates [21], with potential variance reduction [22, 23, 20]; or biased estimates with small variance [24]. However, different algorithms in this space are motivated and derived in seemingly unrelated ways: for example, [21–23] derive code-level implementations within the general context of stochastic computation graphs [25]. On the other hand, [24] derives the estimates by explicitly analyzing certain terms with potentially high variance, which naturally produces bias in the final estimate. Due to apparently distinct ways of deriving estimates, it is not immediately clear how all such methods are related, and whether there could be other alternatives.

**Central contribution.** We present a unified framework for estimating higher-order derivatives of value functions, based on the concept of off-policy evaluation. The main insights are summarized in Table 1, where most aforementioned prior work can be interpreted as special cases of our framework. Our framework has a few advantages: (1) it conceptually unifies a few seemingly unrelated prior methods; (2) it elucidates the bias and variance trade-off of the estimates; (3) it naturally produces new methods based on Taylor expansions of value functions [29].

After a brief background introduction on meta-RL in Section 2, we will discuss the above aspects in detail in Section 3. From an implementation perspective, we will show in Section 4.1 that both the general framework and the new method can be conveniently implemented in auto-differentiation libraries [30, 19], making it amenable in a practical setup. Finally in Section 5, we validate important claims based on this framework with experimental insights.

## 2 Background

We begin with the notation and background on RL and meta-RL.

### 2.1 Task-based reinforcement learning

Consider a Markov decision process (MDP) with state space $\mathcal{X}$ and action space $\mathcal{A}$. At time $t \geq 0$, the agent takes action $a_t \in \mathcal{A}$ in state $x_t \in \mathcal{X}$, receives a reward $r_t$ and transitions to a next state $x_{t+1} \sim p(\cdot|x_t, a_t)$. Without loss of generality, we assume a single starting state $x_0$. Here, we assume the reward $r_t = r(x_t, a_t, g)$ to be a deterministic function of state-action pair $(x_t, a_t)$ and the task variable $g \in \mathcal{G}$. The task variable $g \sim p_{\mathcal{G}}$ is resampled for every episode. For example, $x_t \in \mathcal{X}$ is the sensory space of a running robot, $a_t \in \mathcal{A}$ is the control, $g$ is the episodic target direction in which to run and $r(x_t, a_t, g)$ is the speed in direction $g$. A policy $\pi : \mathcal{X} \to \mathcal{P}(\mathcal{A})$ specifies a distribution over actions at each state. For convenience, we define

the value function $V^\pi(x, g) := \mathbb{E}_\pi \left[ \sum_{t=0}^\infty \gamma^t r(x_t, a_t, g) \mid x_0 = x \right]$ and Q-function $Q^\pi(x, a, g) := \mathbb{E}_\pi \left[ \sum_{t=0}^\infty \gamma^t r(x_t, a_t, g) \mid x_0 = x, a_0 = a \right]$. Without loss of generality, we assume the policy to be smoothly parameterized as $\pi_\theta$ with parameter $\theta \in \mathbb{R}^D$. We further assume that the MDPs terminate within a finite horizon of $H < \infty$ under any policy.

## 2.2 Gradient-based meta-reinforcement learning

The motivation of meta-RL is to identify a policy $\pi_\theta$ such that given a task $g$, after updating the parameter from $\theta$ to $\theta'$ with a parameter update computed under rewards $r(x, a, g)$, the resulting policy $\pi_{\theta'}$ performs well. Formally, define $U(\theta, g) \in \mathbb{R}^D$ as a parameter update to $\theta$, for example, one policy gradient ascent step. Model-agnostic meta-learning (MAML) [4] formulates meta-RL as optimizing the value function at the updated policy $\mathbb{E}_{g \sim p_\mathcal{G}} [V^{\pi_{\theta'}}(x, g)]$ where $\theta' = \theta + U(\theta, g)$ is the updated policy. The optimization is with respect to the initial policy parameter $\theta$. The aim is to find $\theta$ such that it entails fast learning (or adaptation) to the environment, through the *inner loop* update operator $U(\theta, g)$, that leads to high-performing updated policy $\theta'$. Consider the following problem,

$$\max_\theta \ F(\theta) := \mathbb{E}_{g \sim p_\mathcal{G}} \left[ V^{\pi_{\theta'}}(x_0, g) \right], \ \theta' = \theta + U(\theta, g). \tag{1}$$

We can decompose the meta-gradient into two terms based on the chain rule,

$$\nabla_\theta F(\theta) = \mathbb{E}_{g \sim p_\mathcal{G}} \left[ \nabla_\theta F(\theta, g) \right] := \mathbb{E}_{g \sim p_\mathcal{G}} \left[ \nabla_\theta \theta' \nabla_{\theta'} V^{\pi_{\theta'}}(x_0, g) \right] \tag{2}$$

where $\nabla_\theta \theta' = I + \nabla_\theta U(\theta, g) \in \mathbb{R}^{D \times D}$ is a matrix, and $\nabla_{\theta'} V^{\pi_{\theta'}}(x_0, g)$ is the vanilla policy gradient evaluated at the updated parameter $\theta'$ [31]. A straightforward way to optimizing Eqn (1) is to carry out the *outer loop* update $\theta \leftarrow \theta + \alpha \nabla_\theta F(\theta)$ with learning rate $\alpha > 0$.

**Policy gradient update.** Following MAML [4], we focus on policy gradient update where $U(\theta, g) = \eta \nabla_\theta V^{\pi_\theta}(x_0, g)$ for a fixed step size $\eta > 0$. The matrix $\nabla_\theta U(\theta, g) = \eta \nabla_\theta^2 V^{\pi_\theta}(x_0, g)$ is the Hessian of the value function with respect to policy parameters; henceforth, we define $H_\theta(x_0, g) := \nabla_\theta^2 V^{\pi_\theta}(x_0, g) \in \mathbb{R}^{D \times D}$.

**Estimating meta-gradients.** Practical algorithms construct stochastic estimates of the meta-gradients using a finite number of samples. In Eqn (2), we decompose the meta-gradients into the multiplication of a Hessian matrix with a policy gradient vector. Given a fixed task variable $g$, a common practice of prior work is to construct the gradient estimate as the product of the Hessian estimate and the policy gradient estimate $\widehat{\nabla} F(\theta, g) = (I + \eta \widehat{H}_\theta(x_0, g)) \widehat{\nabla}_{\theta'} V^{\pi_{\theta'}}(x_0, g)$; see, e.g., [4, 32, 33, 24, 21–23, 20]. See Algorithm 1 for the full pseudocode for estimating meta-gradients by sampling multiple tasks. Since there is a large literature on constructing accurate estimates to policy gradient (e.g., actor-critic algorithms use baselines for variance reduction [34]), the main challenge consists in estimating the Hessian matrix accurately.

**Bias of common plug-in estimators.** Common practices in meta-RL algorithms rely on the premise that if both Hessian and policy gradient estimates are unbiased, then the meta-gradient estimate is unbiased too.

$$\mathbb{E}\left[\widehat{H}_\theta(x_0, g)\right] = H_\theta(x_0, g), \mathbb{E}\left[\widehat{\nabla}_{\theta'} V^{\pi_{\theta'}}(x_0, g)\right] = \nabla_{\theta'} V^{\pi_{\theta'}}(x_0, g) \Rightarrow \mathbb{E}\left[\widehat{\nabla}_\theta F(\theta, g)\right] = \nabla_\theta F(\theta, g).$$

Unfortunately, this is not true. This is because the two estimates are in general correlated when the sample size is finite, leading to the bias of the overall estimate. We provide further discussions in Appendix A. For the rest of the paper, we follow practices of prior work and focus on the properties of Hessian estimates, leaving a more proper treatment of this bias to future work.

## 3 Deriving Hessian estimates with off-policy evaluation

Since the meta-gradient estimates are computed by averaging over task variables $g \sim p_\mathcal{G}$, in the following, we focus on Hessian estimates at a single state and task variable $H_\theta(x, g)$ with a fixed $g$. In this section, we also drop the dependency of the value function on $g$, such that, e.g., $V^{\pi_\theta}(x_0) \equiv V^{\pi_\theta}(x_0, g)$ and $Q(x, a, g) \equiv Q(x, a)$.

### 3.1 Off-policy evaluation: maintaining higher-order dependencies on parameters

We assume access to data $(x_t, a_t, r_t)_{t=0}^\infty$ generated under a behavior policy $\mu$. Off-policy evaluation [26] consists in building estimators $\widehat{V}^{\pi_\theta}(x, g)$ using the behavior data such that $\widehat{V}^{\pi_\theta}(x) \approx V^{\pi_\theta}(x)$

**Algorithm 1** Pseudocode for computing meta-gradients for the MAML objective

---
**for** i=1,2...n **do**

    Sample task variable $g_i \sim p_\mathcal{G}$.

    Sample $B$ trajectories under policy $\pi_\theta$; Compute $B$-trajectory policy gradient estimate $\widehat{\nabla}_\theta V^{\pi_\theta}(x_0, g_i)$, update parameter $\theta' = \theta + \widehat{\nabla}_\theta V^{\pi_\theta}(x_0, g_i)$.

    Compute $B$-trajectory Hessian estimate $\widehat{H}_\theta(x_0, g_i)$ and an unbiased policy gradient estimate at $\theta'$, i.e., $\widehat{\nabla}_{\theta'} V^{\pi_\theta}(x_0, g)$.

    Compute the $i$-th meta-gradient estimate $\widehat{\nabla} F(\theta, g_i) = \left( I + \eta \widehat{H}_\theta(x_0, g_i) \right) \widehat{\nabla}_{\theta'} V^{\pi_{\theta'}}(x_0, g_i)$.

**end for**

Output averaged meta-gradient estimate $\frac{1}{n} \sum_{i=1}^n \widehat{\nabla}_\theta F(\theta, g_i)$.

---

for a range of target policies $\pi_\theta$. Note that the estimate $\widehat{V}^{\pi_\theta}(x)$ is a random variable depending on $(x_t, a_t, r_t)_{t=0}^\infty$, it is also a function of $\theta$. The approximation $\widehat{V}^{\pi_\theta}(x) \approx V^{\pi_\theta}(x)$ implies that $\widehat{V}^{\pi_\theta}(x)$ is indicative of how the value function $V^{\pi_\theta}(x)$ depends on $\theta$, and hence maintains the higher-order dependencies on $\theta$. Throughout, we assume $\pi_\theta(a|x) > 0, \mu(a|x) > 0$ for all $(x, a) \in \mathcal{X} \times \mathcal{A}$.

**Example: step-wise importance sampling (IS) estimate.** As a concrete example, consider the unbiased step-wise IS estimate $\widehat{V}_{\text{IS}}^{\pi_\theta}(x_0) = \sum_{t=0}^\infty \gamma^t \left( \Pi_{s \leq t} \rho_s^\theta \right) r_t$ where $\rho_s^\theta := \pi_\theta(a_s|x_s)/\mu(a_s|x_s)$. Since the value function $V^{\pi_\theta}(x)$ is in general a highly non-linear function of $\theta$ (see discussions in [35]) we see that $\widehat{V}_{\text{IS}}^{\pi_\theta}(x_0)$ retains such dependencies via the sum of product of IS ratios.

### 3.2 Warming up: deriving unbiased estimates with variance reduction

We start with a general result based on the intuition above: given an estimate $\widehat{V}^{\pi_\theta}(x)$ to $V^{\pi_\theta}(x)$, we can directly use the $m^{\text{th}}$-order derivative $\nabla_\theta^m \widehat{V}^{\pi_\theta}(x) \in \mathbb{R}^{D^K}$ as an estimate to $\nabla_\theta^K V^{\pi_\theta}(x)$. We introduce two assumptions: **(A.1)** $\widehat{V}^{\pi_\theta}(x)$ is $m^{\text{th}}$-order differentiable w.r.t. $\theta$ almost surely. **(A.2)** $\left\| \nabla_\theta^m \widehat{V}^{\pi_\theta}(x) \right\|_\infty < M$ for some constant $M$ for the order $m$ of interest. These assumptions are fairly mild; see further details in Appendix C. The following result applies to general unbiased off-policy evaluation estimates.

**Proposition 3.1.** Assume **(A.1)** and **(A.2)** are satisfied. Further assume we have an estimator $\widehat{V}^{\pi_\theta}(x)$ which is unbiased ($\mathbb{E}_\mu \left[ \widehat{V}^{\pi_{\theta'}}(x) \right] = V^{\pi_{\theta'}}(x)$) for all $\theta' \in N(\theta)$ where $N(\theta)$ is some open set that contains $\theta$. Under some additional mild conditions, the $m^{\text{th}}$-order derivative of the estimate $\nabla_\theta^m \widehat{V}^{\pi_\theta}(x_0)$ are unbiased estimates to the $m^{\text{th}}$-order derivative of the value function $\mathbb{E}_\mu \left[ \nabla_\theta^m \widehat{V}^{\pi_\theta}(x) \right] = \nabla_\theta^m V^{\pi_\theta}(x)$ for $m \geq 1$.

**Doubly-robust estimates.** As a special case, we describe the doubly-robust (DR) off-policy evaluation estimator [36, 27, 37]. Assume we have access to a state-action dependent critic $Q(x, a, g)$, and we use the notation $Q(x, \pi(x)) := \sum_a Q(x, a)\pi(a|x)$. The DR estimate is defined recursively as follows,

$$\widehat{V}_{\text{DR}}^{\pi_\theta}(x_t) = Q(x_t, \pi_\theta(x_t)) + \rho_t^\theta \left( r_t + \gamma \widehat{V}_{\text{DR}}^{\pi_\theta}(x_{t+1}) - Q(x_t, a_t) \right). \tag{3}$$

The DR estimate is unbiased for all $\pi_\theta$ and subsumes the step-wise IS estimate as a special case when $Q(x, a) \equiv 0$. If the critic is properly chosen, e.g., $Q(x, a) \approx Q^{\pi_\theta}(x, a)$, it can lead to significant variance reduction compared to $\widehat{V}_{\text{IS}}^{\pi_\theta}(x_0)$. By directly differentiating the estimate $\nabla_\theta^m \widehat{V}_{\text{DR}}^{\pi_\theta}(x)$, we derive estimators for higher-order derivatives of the value function; the result for the gradient in Proposition 3.2 was shown in [38].

**Proposition 3.2.** Define $\pi_t := \pi_\theta(a_t|x_t)$ and let $\delta_t := r_t + \gamma \widehat{V}_{\text{DR}}^{\pi_\theta}(x_{t+1}) - Q(x_t, a_t)$ be the sampled temporal difference error at time $t$. Note that $\nabla_\theta \log \pi_t \in \mathbb{R}^D$ and $\nabla_\theta^2 \log \pi_t \in \mathbb{R}^{D \times D}$. The estimates

of higher-order derivatives can be deduced recursively, and in particular for $m = 1, 2$,

$$\nabla_\theta \widehat{V}_{\text{DR}}^{\pi_\theta}(x_t) = \nabla_\theta Q(x_t, \pi_\theta(x_t)) + \rho_t^\theta \delta_t \nabla_\theta \log \pi_t + \gamma \rho_t^\theta \nabla_\theta \widehat{V}_{\text{DR}}^{\pi_\theta}(x_{t+1}), \tag{4}$$

$$\nabla_\theta^2 \widehat{V}_{\text{DR}}^{\pi_\theta}(x_t) = \rho_t^\theta \delta_t \left( \nabla_\theta^2 \log \pi_t + \nabla_\theta \log \pi_t \nabla_\theta \log \pi_t^T \right) + \gamma \rho_t^\theta \nabla_\theta \widehat{V}_{\text{DR}}^{\pi_\theta}(x_t) \nabla_\theta \log \pi_t^T$$
$$+ \gamma \rho_t^\theta \nabla_\theta \log \pi_t \nabla_\theta \widehat{V}_{\text{DR}}^{\pi_\theta}(x_t)^T + \nabla_\theta^2 Q(x_t, \pi_\theta(x_t)) + \gamma \rho_t^\theta \nabla_\theta^2 \widehat{V}_{\text{DR}}^{\pi_\theta}(x_{t+1}). \tag{5}$$

**Bias and variance of Hessian estimates.** Proposition 3.1 implies that $\nabla_\theta \widehat{V}_{\pi_\theta}(x_0)$ and $\nabla_\theta^2 \widehat{V}_{\pi_\theta}(x_0)$ are both unbiased. To analyze the variance, we start with $m = 1$: note that when on-policy $\mu = \pi_t$ [16], $\nabla_\theta V_{\text{DR}}^{\pi_\theta}(x_0)$ recovers a form of gradient estimates similar to actor-critic policy gradient with action-dependent baselines [39–41]; when $Q(x, a)$ is only state dependent, $\nabla_\theta \widehat{V}_{\text{DR}}^{\pi_\theta}(x)$ recovers the common policy gradient estimate with state-dependent baselines [34]. As such, the estimates are computed with potential variance reduction due to the critic. Previously, [38] started with the DR estimate and derived a more general result for the on-policy first-order case. For the Hessian estimate, we expect a similar effect of variance reduction as shown in experiments.

**Recovering prior work on estimates to higher-order derivatives.** When applied to meta-RL, DiCE [21] and its follow-up variants [22, 42] can be seen as special cases of $\nabla_\theta^2 \widehat{V}_{\text{DR}}^{\pi_\theta}(x)$ with different choices of the critic $Q$ when evaluated on-policy $\mu = \pi_\theta$. See Table 1 for the correspondence between prior work and their equivalent formulations under the framework of off-policy evaluation. We will discuss detailed pseudocode in Section 4.1. See also Appendix F for more details.

### 3.3 Trading-off bias and variance with Taylor expansions

Starting from unbiased off-policy evaluation estimates $\widehat{V}^{\pi_\theta}(x)$, we can directly construct unbiased estimates to higher-order derivatives by differentiating the original estimate to obtain $\nabla_\theta^m \widehat{V}^{\pi_\theta}(x)$. However, unbiased estimates can have large variance. Though it is possible to reduce variance through the critic, as we will show experimentally, this is not enough to counter the high variance due to products of IS ratios. This leads us to consider trading off bias with variance [43].

Since we postulate that the products of IS ratios lead to high variance, we might seek to control the number of IS ratios in the estimate. We briefly introduce Taylor expansion policy optimization (TayPO) [29], a natural framework to control for the number of IS ratios in the value estimate.

**Taylor expansions of value functions.** Consider the value function $V^{\pi_\theta}(x_0)$ as a function of $\pi_\theta$. Using the idea of Taylor expansions, we can express $V^{\pi_\theta}(x_0)$ as a sum of polynomials of $\pi_\theta - \mu$. We start by defining the $K^{\text{th}}$-order increment as $U_0^{\pi_\theta}(x_0) = V^\mu(x_0)$, which does not contain any IS ratio (zeroth order); and for $K \geq 1$,

$$U_K^{\pi_\theta}(x_0) := \mathbb{E}_\mu \bigg[ \underbrace{\sum_{t_1=0}^\infty \sum_{t_2=t_1+1}^\infty \cdots \sum_{t_K=t_{K-1}+1}^\infty \gamma^{t_K} \left( \Pi_{i=1}^K (\rho_{t_i}^\theta - 1) \right) Q^\mu(x_{t_K}, a_{t_K})}_{\widehat{U}_K^{\pi_\theta}(x_0)} \bigg]. \tag{6}$$

Intuitively, the $K^{\text{th}}$-order increment only contains product of $K$ IS ratios. Equation (6) also yields a natural sample-based estimate $\widehat{U}_K^{\pi_\theta}(x_0)$, which we will discuss later. The $K^{\text{th}}$-order Taylor expansion is defined as the partial sum of increments $V_K^{\pi_\theta}(x_0) := \sum_{k=0}^K U_k^{\pi_\theta}(x_0)$. Since $V_K^{\pi_\theta}(x_0)$ consists of products of up to $K$ IS ratios, it is effectively the $K^{\text{th}}$-order Taylor expansion of the value function. The properties are summarized as follows.

**Proposition 3.3.** (Adapted from Theorem 2 of [29].) Define $\|\pi - \mu\|_1 := \max_x \sum_a |\pi(a|x) - \mu(a|x)|$. Let $C$ be a constant and $\varepsilon = \frac{1-\gamma}{\gamma}$. Then the following holds for all $K \geq 0$,

$$V^{\pi_\theta}(x_0, g) = \underbrace{V_K^{\pi_\theta}(x_0, g)}_{K\text{-th order expansion}} + \underbrace{C(\|\pi_\theta - \mu\|_1 / \varepsilon)^{K+1}}_{\text{residual}}, \tag{7}$$

If $\|\pi_\theta - \mu\|_1 < \varepsilon$, then $V^{\pi_\theta}(x_0) = \lim_{K\to\infty} V_K^{\pi_\theta}(x_0) = \mathbb{E}_\mu \left[ \sum_{k=0}^\infty U_k^{\pi_\theta}(x_0) \right]$.

**Sample-based estimates to Taylor expansions of value functions.** As shown in Equation (6), $\widehat{U}_K^{\pi_\theta}(x_0)$ is an unbiased estimate to $U_K^{\pi_\theta}(x_0)$. We can naturally define the sample-based estimate to the $K^{\text{th}}$-order Taylor expansion, called the TayPO-$K$ estimate,

$$\widehat{V}_K^{\pi_\theta}(x_0) := \sum_{k=0}^{K} \widehat{U}_k^{\pi_\theta}(x_0). \tag{8}$$

The expression of $\widehat{U}^{\pi_\theta}(x_0)$ contains $O(T^K)$ terms if the trajectory is of length $T$. Please refer to Appendix E for further details on computing the estimates in linear time $O(T)$ with sub-sampling. Note that $\widehat{V}_K^{\pi_\theta}(x_0)$ is a sample-based estimate whose bias against the value function is controlled by the residual term which decays exponentially when $\pi_\theta$ and $\mu$ are close. Similar to how we derived the unbiased estimate $\nabla_\theta^m \widehat{V}_{\text{DR}}^{\pi_\theta}(x)$, we can differentiate through the TayPO-$K$ value estimate to produce estimates to higher-order derivatives $\nabla_\theta^m \widehat{V}_K^{\pi_\theta}(x_0)$.

**Bias and variance of Hessian estimates.** TayPO-$K$ trades-off bias and variance with choices of $K$. To understand the variance, note that TayPO-$K$ limits the number multiplicative IS ratios to be $K$. Though it is difficult to compute the variance, we argue that the variance generally increases with $K$ as the number of IS ratios increase [26, 27, 44]. We characterize the bias of TayPO-$K$ as follows.

**Proposition 3.4.** Assume **(A.1)** and **(A.2)** hold. Also assume $\|\pi_\theta - \mu\|_1 \leq \varepsilon = (1 - \gamma)/\gamma$. For any tensor $x$, define $\|x\|_\infty := \max_i |x[i]|$. The $K^{\text{th}}$-order TayPO objective produces the following bias in estimating high-order derivatives,

$$\left\| \mathbb{E}_\mu \left[ \nabla_\theta^m \widehat{V}_K^{\pi_\theta} \right](x_0) - \nabla_\theta^m V^{\pi_\theta}(x_0) \right\|_\infty \leq \sum_{k=K+1}^{\infty} \left\| \nabla_\theta^m U_k^{\pi_\theta}(x_0) \right\|_\infty. \tag{9}$$

Hence the upper bound for the bias decreases as $K$ increases. Importantly, when on-policy $\mu = \pi_\theta$, the $K^{\text{th}}$-order TayPO objective preserves up to $K^{\text{th}}$-order derivatives for any $K \geq 0$,

$$\mathbb{E}_\mu \left[ \nabla_\theta^m \widehat{V}_K^{\pi_\theta}(x_0) \right] = \nabla_\theta^m V^{\pi_\theta}(x_0), \forall m \leq K. \tag{10}$$

Though IS ratios $\rho_t^\theta$ evaluate to 1 when on-policy, they maintain the parameter dependencies in differentiations. As such, higher-order expansions contains products of IS ratios of higher orders, and maintains the high-order dependencies on parameters more accurately. There is a clear trade-off between bias and variance mediated by $K$. When $K$ increases, the higher-order derivatives are maintained more accurately in expectation, leading to less bias. However, the variance increases too.

**Recovering prior work as special cases.** Recently, [24] proposed a low variance curvature (LVC) Hessian estimate. This estimate is equivalent to the TayPO-$K$ estimate with $K = 1$. As also noted by [24], their objective function bears similarities to first-order policy search algorithms [28, 45, 46], which have in fact been interpreted as first-order special cases of $\nabla_\theta \widehat{V}_K^{\pi_\theta}(x)$ with $K = 1$ [29]. Importantly, based on Proposition 3.4, the LVC estimate only maintains the first-order dependency perfectly but introduces bias when approximating the Hessian, even when on-policy.

**Limitations.** Though the above framework interprets a large number of prior methods as special cases, it has some limitations. For example, the derivation of Hessian estimates based on the DR estimate (Proposition 3.2) involves estimates of the value function $\widehat{V}_{\text{DR}}^{\pi_\theta}(x_t)$. In practice, when near on-policy $\pi_\theta \approx \mu$, one might replace the DR estimate $\widehat{V}_{\text{DR}}^{\pi_\theta}$ by other value function estimate, such as plain cumulative sum of returns or TD($\lambda$), $\widehat{V}_{\text{TD}(\lambda)}^{\pi_\theta}$ [22, 23] in Eqn (4). As such, the practical implementation might not strictly adhere to the conceptual framework. In addition, TMAML [20] is not incorporated as part of this framework: we show in Appendix B that the control variate introduced by TMAML in fact biases the overall estimate.

## 4  From Hessian estimates to meta-gradient estimates

A practical desideratum for meta-gradient estimates is that it can be implemented in a scalable way using auto-differentiation frameworks [30, 19]. Below, we discuss how this can be achieved.

## 4.1 Auto-differentiating off-policy evaluation estimates for Hessian estimates

In practice, we seek Hessian estimates that could be implemented with the help of an established framework, such as auto-differentiation libraries [30, 19]. Now we discuss how to conveniently implement ideas discussed in the previous section.

---

**Algorithm 2** Example: an off-policy evaluation subroutine for computing the DR estimate

---

**Require: Inputs**: Trajectory $(x_t, a_t, r_t)_{t=0}^T$, target policy $\pi_\theta$, behavior policy $\mu$, (optional) critic $Q$.

    Initialize $\widehat{V} = Q(x_T, \pi_\theta(x_T), g)$.

    **for** $t = T - 1, \ldots 0$ **do**

        Compute IS ratio $\rho_t^\theta = \pi_\theta(a_t|x_t)/\mu(a_t|x_t)$.

        Recursion: $\widehat{V} \leftarrow Q(x_t, \pi_\theta(a_t), g) + \gamma\rho_t^\theta(r_t + \gamma Q(x_{t+1}, \pi_\theta(x_{t+1}), g) - Q(x_t, a_t)) + \gamma\rho_t^\theta\widehat{V}$.

    **end for**

    Output $\widehat{V}$ as an estimate to $V^{\pi_\theta}(x_0, g)$.

---

**Auto-differentiating the estimates.** We can abstract the off-policy evaluation as a function $\text{eval}(D, \theta)$ that takes in some data $D$ and parameter $\theta$, and outputs an estimate for $V^{\pi_\theta}(x_0, g)$. In particular, $D$ includes the trajectories and $\theta$ is input via the policy $\pi_\theta$. As an example, Algorithm 2 shows that for the doubly-robust estimator, the dependency of the estimator on $\theta$ is built through the recursive computation by $\text{eval}(D, \theta)$. In fact, if we implement Algorithm 2 with an auto-differentiation framework (for example, Tensorflow or PyTorch [17–19]), the higher-order dependency of $\widehat{V}$ on $\theta$ is maintained through the computation graph. We can compute the Hessian estimate by directly differentiating through the function output $\nabla_\theta^K \widehat{V} = \nabla_\theta^K \text{eval}(D, \theta) \approx \nabla_\theta^K V^{\pi_\theta}(x_0, g)$. In Appendix F we show how the estimates could be conveniently implemented, as in many deep RL agents (see, e.g., [44, 5, 47, 48]). We also show concrete ways to compute estimates with TayPO-$K$ based on [29].

## 4.2 Practical implementations of meta-gradient estimates

In Equation (2), we write the meta-gradient estimate as a product between an Hessian estimate and a policy gradient. In practice, the meta-gradient estimate is computed via Hessian-vector products to avoid explicitly computing the Hessian estimate of size $D^2$. As such, the meta-gradient estimate could be computed by auto-differentiating through a scalar objective. See Appendix F for details.

**Bias and variance of meta-gradient estimates.** Intuitively, the bias and variance of the Hessian estimates translate into bias and variance of the downstream meta-gradient estimates. Prior work has showed that low variance of meta-gradient estimates lead to faster convergence [49, 50]. However, it is not clear how the bias (such as bias introduced by the Hessian estimates, or the bias due to correlated estimates) theoretically impacts the convergence. We will study empirically the effect of bias and variance in experiments, and leave further theoretical study to future work.

# 5 Experiments

We now carry out several empirical studies to complement the framework developed above. In Section 5.1, we use a tabular example to investigate the bias and variance trade-offs of various estimates, to assess the validity of our theoretical insights. We choose a tabular example because it is straightforward to compute exact higher-order derivatives of value functions and make comparison. In Section 5.2.1 and Section 5.2.2, we apply the new second-order estimate in high-dimensional meta-RL experiments, to assess the potential performance gains in a more practical setup. Though we can compute TayPO-$K$ order estimates for general $K$, we focus on $K \leq 2$ in experiments. Below, we also address TayPO-1 and TayPO-2 estimates as the first and second-order estimates respectively.

## 5.1 Investigating the bias and variance trade-off of different estimates

We study the bias and variance trade-off of various estimates using a tabular example. We consider random MDPs with $|\mathcal{X}| = 10, |\mathcal{A}| = 5$. The transition matrix of the MDPs are sampled from a Dirichlet distribution. See Appendix G for further details. The policy $\pi_\theta$ is parameterized as $\pi_\theta(a|x) = \exp(\theta(x, a))/\sum_b \exp(\theta(x, b))$. The behavior policy $\mu$ is uniform and $\theta$ is initialized so that $\theta(x, a) = \log \pi(a|x)$ where $\pi = (1 - \varepsilon)\mu + \varepsilon\pi_d$ for some deterministic policy $\pi_d$ and parameter $\varepsilon \in [0, 1]$. The hyper-parameter $\varepsilon$ measures the off-policyness. In this example, there is no task variable. As performance metrics, we measure the component-wise correlation between the true derivatives $\nabla_\theta^K V^{\pi_\theta}(x_0)$ (computed via an oracle) where $x_0$ is a fixed starting state, and its estimates

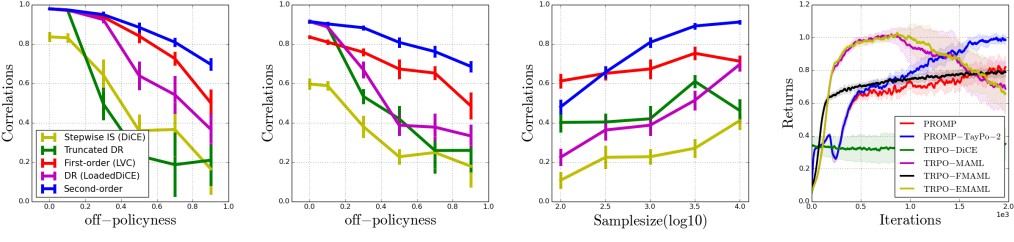

(a) Gradient - off-policy    (b) Hessian - off-policy    (c) Hessian - sample size    (d) 2-$D$ control: training

Figure 1: Fig (a)-(b): performance measure as a function of off-policyness measured by $\varepsilon = \|\pi_\theta - \mu\|_1$. (a) shows results for gradient estimation and (b) shows Hessians. Plots show the accuracy measure between the estimates and the ground truth. Overall, the second-order estimate achieves a better bias and variance trade-off. Fig (c): performance measure as a function of off-policyness measured by sample size $N$ for Hessians. Fig (d): training curves for the 2-$D$ control environment. The second-order estimate is generally more robust. All curves are averaged over 10 runs. The left three plots use share the same legends.

$\nabla_\theta^K \widehat{V}^{\pi_\theta}(x_0)$, as commonly used in prior work [21–23]. The estimates are averaged across $N$ samples. We study the effect of different choices of the off-policy estimate $\widehat{V}^{\pi_\theta}$, as a function of off-policyness $\varepsilon$ and sample size $N$. We report results with mean $\pm$ standard error over 10 seeds.

**Effect of off-policyness.** In Figure 1(a) and (b), we let $N = 1000$ and show the performance as a function of $\varepsilon$. When $\varepsilon$ increases (more off-policy), the accuracy measures of most estimates decrease. This is because off-policyness generally increases both bias and variance (large IS ratios $\rho^\theta$). At this level of sample size, the performance of the second-order estimate degrades more slowly than other estimates, making it the dominating estimate across all values of $\varepsilon$. We also include truncated DR estimate using $\min(\rho_t^\theta, \bar{\rho})$ as a baseline inspired by V-trace [44, 47]. The truncation is motivated by controlling the variance of the overall estimate. However, the estimate is heavily biased and does not perform well unless $\pi_\theta \approx \mu$. See Appendix F for more.

Consider the case when $\varepsilon = 0$ and the setup is on-policy. When estimating the policy gradient, almost all estimates converge to the optimal level of accuracy, except for the step-wise IS estimate, where the variance still renders the performance sub-optimal. However, when estimating the Hessian matrix, the first-order estimate converges to a lower accuracy than both second-order estimate and DR estimate. This validates our theoretical analysis, as both the second-order estimate and DR estimate are unbiased when on-policy.

**Effect of sample size.** Figure 1(c) shows the accuracy measures as a function of sample size $N$ when fixing $\varepsilon = 0.5$. Note that since sample sizes directly influence the variance, the results show the variance properties of different estimates. When $N$ is small, the first-order estimate dominates due to smaller variance; however, when $N$ increases, the first-order estimate is surpassed by the second-order estimate, due to higher bias. For more results and ablation on high-dimensional environments, see Appendix G.

## 5.2 High-dimensional meta-RL problems

Next, we study the practical gains entailed by the second-order estimate in high-dimensional meta-RL problems. We first introduce a few important algorithmic baselines, and how the second-order estimate is incorporated into a meta-RL algorithm.

**Baseline algorithms.** All baseline algorithms use plain stochastic gradient ascent as the inner loop optimizer: $\theta' = \theta + \eta \nabla_\theta \widehat{V}^{\pi_\theta}(x_0, g)$ where $g \sim p_{\mathcal{G}}$ is a sampled goal and $\widehat{V}^{\pi_\theta}(x_0, g)$ is a sample-based estimate of policy gradients averaged over $n$ trajectories. Different algorithms differ in how the inner loop loss is implemented, such that auto-differentiation produces different Hessian estimates: these include TRPO-DiCE [21–23], TRPO-MAML [4], TRPO-FMAML [4], TRPO-EMAML [32, 33]. Please refer to Appendix G for further details. Note despite the name, TRPO-DiCE baseline uses DR estimate to estimate Hessians. We implement the second-order

estimate using the proximal meta policy search (PROMP) [24] as the base algorithm. By default, PROMP uses the first-order estimate. Our new algorithm is named PROMP-TayPO-2.

### 5.2.1 Continuous control in 2D environments

**Environment.** We consider a simple 2-D navigation task introduced in [24]. The state $x_t$ is the coordinate of a ball placed inside a room, the action $a_t$ is the direction in which to push the ball. The goal $g \in \mathbb{R}^4$ is an one-hot encoding of which corner of the room contains positive rewards. With 3 adaptation steps, the agent should ideally navigate to the desired corner indicated by $g$.

**Training performance.** In Figure 1(d), we show the performance curves of various baseline algorithms. Though MAML and EMAML learns quickly during the initial phase of training, they ultimately become unstable. TRPO-DiCE generally underperforms other methods potentially due to bias. On the other hand, FMAML and PROMP both reduce variance at the cost of bias, but they both achieve a slightly lower level of performance compared to PROMP-TayPO-2. Overall, PROMP-TAyPO-2 achieves much more stable training curves compared to others, potentially owing to the better bias and variance trade-off in the Hessian estimates.

### 5.2.2 Large scale locomotion experiments

**Environments.** We consider the set of meta-RL tasks based on simulated locomotion in MuJoCo [51]. Across these tasks, the states $x_t$ consist of robotic sensory inputs, the actions $a_t$ are torque controls applied to the robots. The task $g$ is defined per environment: for example, in random goal environment, $g \in \mathbb{R}^2$ is a random 2-d goal location that the robot should aim to reach.

**Experiment setup.** We adapt the open source code base by [24] and adopt exactly the same experimental setup as [24]. At each iteration, the agent samples $n = 40$ task variables. For each task, the agent carries out $K = 1$ adaptation computed based on $B = 20$ trajectories sampled from the environment, each of length $T = 100$. See Appendix G for further details on the architecture and other hyper-parameters. We report averaged results with mean $\pm$ std over 10 seeds.

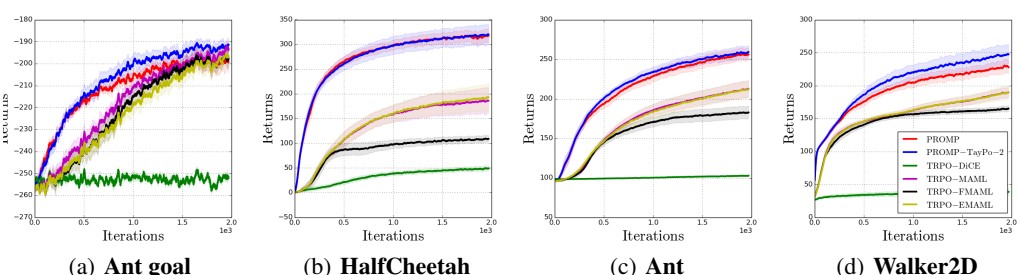

|  |  |  |  |
|---|---|---|---|
| (a) **Ant goal** | (b) **HalfCheetah** | (c) **Ant** | (d) **Walker2D** |

Figure 2: Comparison of baselines over a range of simulated locomotion tasks. For task (b)-(d), the goal space consists of 2-d random direction in which the robot should run to obtain positive rewards. For task (a), the goal space is a 2-d location on the plane. Each curve shows the mean $\pm$ std across 5 seeds. Overall, the second-order estimate achieves marginal gains over the first-order estimate.

**Results.** The training performance of different algorithmic baselines are shown in Figure 2. Comparing TRPO-DiCE, TRPO-MAML and PROMP: we see that the results are compatible with those reported in [24], where PROMP outperforms TRPO-MAML, while TRPO-DiCE generally performs the worst potentially due to high variance in the gradient estimates. As a side observation, TRPO-FMAML generally underperforms TRPO-MAML, which implies the necessity of carrying out approximations to the Hessian matrix beyond the identity matrix. PROMP-TayPO-2 slightly outperforms PROMP in a few occasions, where the new algorithm achieves slightly faster learning speed and sometimes higher final performance. However, overall, we see that the empirical gains are marginal. This implies that under the default setup of these meta-RL experiments, the variance might be a major factor in gradient estimates, and the first-order estimate is near optimal compared to other estimates. See Appendix G for additional experiments.

## 6 Conclusion

We have unified a number of important prior work on meta-gradient estimations for model-agnostic meta-RL. Our analysis entails the derivations of prior methods based on the unifying framework of

differentiating through off-policy evaluation estimates. This framework provides a principled way to reason about the bias and variance in the higher-order derivative estimates of value functions, and opens the door to a new family of estimates based on novel off-policy evaluation estimates. As an important example, we have theoretically and empirically studied the properties of the family of TayPO-based estimates. It is worth noting that this framework further suggests any future advances in off-policy evaluation could be conveniently imported into potential improvements in meta-gradient estimates. As future work, we hope to see the applications of such principled estimates in broader meta-RL applications.

**Acknowledgement.** The authors thank David Abel for reviewing an early draft of this work and providing very useful feedback. Yunhao and Tadashi are thankful for the Scientific Computation and Data Analysis section at the Okinawa Institute of Science and Technology (OIST), which maintains a cluster we used for many of our experiments.

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
