## A Bias with practical estimations of meta-gradients

Recall the MAML objective defined in Eqn (1). The adapted parameter $\theta' = \theta + U(\theta, g)$ is computed with the expected policy gradient update $U(\theta, g) = \nabla_\theta V^{\pi_\theta}(x_0, g)$. This makes it difficult to construct fully unbiased estimates to the MAML gradient. We use a following example to show the intuitions.

Consider a scalar objective $f(x)$ with input $x$. If it is possible to construct unbiased estimate to $x$, i.e., with $X$ such that $\mathbb{E}[X] = x$. It is difficult to construct unbiased estimates to $f(x)$ because $\mathbb{E}[f(X)] \neq f(x)$ unless $f$ is linear.

Conceptually, we can think of the argument $x$ here as the updated parameter resulting from expected updates $\theta' := \theta + \eta\nabla_\theta V^{\pi_\theta}(x_0, g)$. It is convenient to construct unbiased estimate of this parameter because it is convenient to build unbiased estimate to the policy gradient $\nabla_\theta V^{\pi_\theta}(x_0, g)$. However, in our case, to compute the meta RL gradients, we need to evaluate the policy gradient at $\theta'$: $\nabla_{\theta'} V^{\pi_{\theta'}}(x_0, g)$, which is usually an highly non-linear function of $\theta'$. Using the notation of the above scalar objective example, we can construct a vector valued function: $f : \theta \mapsto \nabla_\theta V^{\pi_\theta}(x_0, g)$. Though it is straightforward to construct unbiased estimates $\widehat{\theta'}$ to $\theta'$ with sampled trajectories, it is not easy to estimate $f(\theta')$ in an unbiased way, as a direct plug in $f(\widehat{\theta'})$ would be biased.

## B Proof on the bias of TMAML

Below, we adopt the trajectory-based notation of TMAML [20]. Let $\rho_t^\theta := \frac{\pi_\theta(a_t|x_t)}{\text{sg}(\pi_\theta(a_t|x_t))}$, where the operation $\text{sg}(x)$ removes the dependency of $x$ on parameter $\theta$. In other words, $\nabla_\theta \text{sg}(x) = 0$. It is worth noting that the stop gradient notations are equivalent to the derivation that defines $\rho_t^\theta = \frac{\pi_\theta(a_t|x_t)}{\mu(a_t|x_t)}$ with a fixed behavior policy $\mu$, and later sets $\mu = \pi_\theta$, as done in the main paper. TMAML [20] proposed the following baseline objective in the undiscounted finite horizon case with horizon $H < \infty$,

$$J = \sum_{t=0}^{H-1} \left(1 - \left(\Pi_{s=0}^t \rho_s^\theta\right)\right)(1 - \rho_t^\theta)b(x_t),$$

where $b(x_t)$ is a baseline function that depends on state $x_t$. The major claim from TMAML [20] is that $\mathbb{E}_{\pi_\theta}\left[\nabla_\theta^2 J\right] = 0$, which implies that adding $J$ to the original DICE objective [21] might lead to variance reduction because it serves as a control variate term. However, below we show

$$\mathbb{E}_{\pi_\theta}\left[\nabla_\theta^2 J\right] \neq 0.$$

The proof proceeds in deriving the explicit expression for $\nabla_\theta^2 J$. We derive the same expression as that in the Appendix C of the original paper [20], i.e.,

$$\nabla_\theta^2 J = 2 \sum_{t=0}^{H-1} \nabla_\theta \log \pi_\theta(a_t|x_t) \left(\sum_{s=t}^{H-1} \nabla_\theta \log \pi_\theta(a_s|x_s)b(x_s)\right)^T. \tag{11}$$

Following the proof of [20], it is then straightforward to show that

$$\mathbb{E}_{\pi_\theta}\left[2 \sum_{t=0}^{H-1} \nabla_\theta \log \pi_\theta(a_t|x_t) \left(\sum_{s=t+1}^{H-1} \nabla_\theta \log \pi_\theta(a_s|x_s)b(x_s)\right)^T\right] = 0. \tag{12}$$

However, note the difference between Eqn (11) and Eqn (12) lies in the summation $s = t$ instead of $s = t + 1$. Accounting for this difference, we have

$$\mathbb{E}_{\pi_\theta}\left[\nabla_\theta^2 J\right] = \mathbb{E}_{\pi_\theta}\left[2 \sum_{t=0}^{H-1} \nabla_\theta \log \pi_\theta(a_t|x_t) \left(\nabla_\theta \log \pi_\theta(a_t|x_t)\right)^T b(x_t)\right],$$

which in general does not evaluate to zero. In fact, the bias of $\nabla_\theta^2 J$ is clear if we take the special case $H = 1$. In this case, it is more straightforward to derive

$$\mathbb{E}_{\pi_\theta}\left[\nabla_\theta^2 J\right] = \mathbb{E}_{\pi_\theta}\left[2\nabla_\theta \log \pi_\theta(a_0|x_0) \left(\nabla_\theta \log \pi_\theta(a_0|x_0)\right)^T b(x_0)\right].$$

As a result, we showed that TMAML [20] might achieve variance reduction by introducing baselines to the Hessian estimates of DiCE [21], but at the cost of bias.

## C  Further discussions on the assumptions

In this section, we examine if the assumptions **(A.1)** and **(A.2)** are realistic.

Both assumptions depend on particular functional form of the estimates $\widehat{V}^{\pi_\theta}$. In general, we might assume that the estimates do not explicitly depend on $\theta$. Instead, they depend on $\theta$ via $\pi_\theta$. This involves a two-stage parameterization: $\theta \mapsto \pi_\theta$ and $\pi_\theta \mapsto \widehat{V}^{\pi_\theta}$. The two assumptions **(A.1)** and **(A.2)** can be realized by imposing constraints on these two parameterizations, as well as the off-policyness of $\pi_\theta$ relative to $\mu$, as discussed below.

**Off-policyness.**  In general, we might want to assume the ratios are bounded $\rho_t^\theta < R$ for constant some $R < \infty$. This is a common assumption. In our framework, we usually apply the estimates within a trust region optimization algorithm [45, 46], this naturally proIS duces a bound on the ratios.

**Parameterization** $\theta \mapsto \pi_\theta$**.**  We seek parameterizations where $\nabla_\theta \rho_t^\theta$ are bounded. This can be achieved by bounding $\rho_t^\theta$ and $\nabla_\theta \log \pi_\theta(a|x)$. If we consider a tabular representation with softmax parameterization $\pi(a|x) \propto \exp(\theta(x,a))$. Under this parameterization, we can show $|\nabla_\theta^m \log \pi_\theta(a|x)| < M$ are bounded for all $(x,a)$ and all $\theta$.

**Parameterization** $\pi_\theta \mapsto \widehat{V}^{\pi_\theta}$**.**  We want this parameterization to be sufficiently smooth. In the examples we consider, TayPO-$K$ clearly satisfies this assumption because it is a polynomial in $\pi_\theta$. For DR, this assumption is satisfied when the MDP terminates within a finite horizon of $H < \infty$, such that the estimator contains polynomials of $\pi_\theta$ with order at most $H$.

## D  Proof of results in the main paper

**Proposition 3.1.** Assume **(A.1)** and **(A.2)** are satisfied. Further assume we have an estimator $\widehat{V}^{\pi_\theta}(x)$ which is unbiased $(\mathbb{E}_\mu\left[\widehat{V}^{\pi_{\theta'}}(x)\right] = V^{\pi_{\theta'}}(x))$ for all $\theta' \in N(\theta)$ where $N(\theta)$ is some open set that contains $\theta$. Under some additional mild conditions, the $m^{\text{th}}$-order derivative of the estimate $\nabla_\theta^m \widehat{V}^{\pi_\theta}(x_0)$ are unbiased estimates to the $m^{\text{th}}$-order derivative of the value function $\mathbb{E}_\mu\left[\nabla_\theta^m \widehat{V}^{\pi_\theta}(x)\right] = \nabla_\theta^m V^{\pi_\theta}(x)$ for $m \geq 1$.

*Proof.* The two assumptions along with the unbiasedness of the estimates, allow us to exchange $m^{\text{th}}$-order derivatives and the expectation, and hence leading to the unbiasedness of the derivate estimates. The proof is similar to the exchange techniques used in [38] to show the unbiasedness of the first-order derivatives of DR estimates.

We proceed the argument with induction. Assume we have

$$\mathbb{E}_\mu\left[\nabla_\theta^i \widehat{V}^{\pi_\theta}(x_0, g)\right] = \nabla_\theta^i V^{\pi_\theta}(x_0, g),$$

for some $i$. To define the $(i+1)$-th order derivative, we differentiate further through the $i$-th order derivative. Consider some particular component of the $(i+1)$-th order derivative, obtained by taking the derivative with respect to variable $\theta_L$. We now denote this component of the $(i+1)$-th order derivative as $D_L^{(i+1)}[\theta]$ evaluated at $\theta$. Let $D^{(i)}[\theta]$ denote the $i$-th order derivative (a tensor) evaluated at $\theta$. Also define $e_L \in \mathbb{R}^D$ as the one-hot vector such as its $L$-th component is one. By definition,

$$D_L^{(i+1)} := \lim_{h \to 0} \frac{D^{(i)}[\theta + e_L \cdot h] - D^{(i)}[\theta]}{h},$$

we also denote the unbiased estimate to $D^{(i)}[\theta]$ as $\widehat{D}^{(i)}[\theta]$. The new estimate is

$$\widehat{D}_L^{(i+1)} := \lim_{h \to 0} \frac{\widehat{D}^{(i)}[\theta + e_L \cdot h] - \widehat{D}^{(i)}[\theta]}{h}.$$

Now we seek to establish that $\mathbb{E}_\mu\left[\widehat{D}_L^{(i+1)}\right] = D_L^{(i+1)}$. Note that this is equivalent to showing

$$\mathbb{E}_\mu\left[\lim_{h \to 0} \frac{\widehat{D}^{(i)}[\theta + e_L \cdot h] - \widehat{D}^{(i)}[\theta]}{h}\right] = D_L^{(i+1)}.$$

Note that with the RHS, we can use the definition along with unbiasedness of the $i$-th order derivatives

$$D_L^{(i+1)} = \lim_{h \to 0} \frac{\mathbb{E}_\mu \left[ \widehat{D}^{(i)}[\theta + e_L \cdot h] \right] - \mathbb{E}_\mu \left[ \widehat{D}^{(i)}[\theta] \right]}{h}$$

Combining the new RHS into a single expectation, **(A.1)** entails the application of the mean value theorem,

$$\lim_{h \to 0} \mathbb{E}_\mu \left[ \frac{\widehat{D}^{(i)}[\theta + e_L \cdot h] - \widehat{D}^{(i)}[\theta]}{h} \right] = \lim_{h \to 0} \mathbb{E}_\mu \left[ \widehat{D}^{(i+1)}[\theta + e_L \cdot h \cdot \eta] \right],$$

for some $\eta \in (0,1)$. Due to **(A.2)**, we can use dominated convergence theorem to exchange the limit and the expectation,

$$\lim_{h \to 0} \mathbb{E}_\mu \left[ \widehat{D}^{(i+1)}[\theta + e_L \cdot h \cdot \eta] \right] = \mathbb{E}_\mu \left[ \lim_{h \to 0} \widehat{D}^{(i+1)}[\theta + e_L \cdot h \cdot \eta] \right] = \mathbb{E}_\mu \left[ \widehat{D}^{(i+1)}[\theta] \right].$$

This proves the case for $(i+1)$-th order derivative. The base case holds for $i = 0$ and we have the assumptions hold for all $0 \le i \le m - 1$. This concludes the proof of the theorem.

$\square$

**Proposition 3.2.** Define $\pi_t := \pi_\theta(a_t|x_t)$ and let $\delta_t := r_t + \gamma \widehat{V}_{\text{DR}}^{\pi_\theta}(x_{t+1}) - Q(x_t, a_t)$ be the sampled temporal difference error at time $t$. Note that $\nabla_\theta \log \pi_t \in \mathbb{R}^D$ and $\nabla_\theta^2 \log \pi_t \in \mathbb{R}^{D \times D}$. The estimates of higher-order derivatives can be deduced recursively, and in particular for $m = 1, 2$,

$$\nabla_\theta \widehat{V}_{\text{DR}}^{\pi_\theta}(x_t) = \nabla_\theta Q(x_t, \pi_\theta(x_t)) + \rho_t^\theta \delta_t \nabla_\theta \log \pi_t + \gamma \rho_t^\theta \nabla_\theta \widehat{V}_{\text{DR}}^{\pi_\theta}(x_{t+1}), \tag{4}$$

$$\nabla_\theta^2 \widehat{V}_{\text{DR}}^{\pi_\theta}(x_t) = \rho_t^\theta \delta_t \left( \nabla_\theta^2 \log \pi_t + \nabla_\theta \log \pi_t \nabla_\theta \log \pi_t^T \right) + \gamma \rho_t^\theta \nabla_\theta \widehat{V}_{\text{DR}}^{\pi_\theta}(x_t) \nabla_\theta \log \pi_t^T$$
$$+ \gamma \rho_t^\theta \nabla_\theta \log \pi_t \nabla_\theta \widehat{V}_{\text{DR}}^{\pi_\theta}(x_t)^T + \nabla_\theta^2 Q(x_t, \pi_\theta(x_t)) + \gamma \rho_t^\theta \nabla_\theta^2 \widehat{V}_{\text{DR}}^{\pi_\theta}(x_{t+1}). \tag{5}$$

*Proof.* Starting from the definition of the DR estimate in Eqn (3), which we recall here

$$\widehat{V}_{\text{DR}}^{\pi_\theta}(x_t, g) = Q(x_t, \pi_\theta(x_t), g) + \rho_t^\theta \delta_t + \gamma \rho_t^\theta \left( \widehat{V}_{\text{DR}}^{\pi_\theta}(x_{t+1}, g) - Q(x_{t+1}, \pi_\theta(x_{t+1}), g) \right).$$

Note that both sides of the equations are functions of $\pi_\theta$. Since the DR estimate holds for all $\pi_\theta$ (assuming $\mu$ has larger support than $\pi_\theta$), and both sides are differentiable functions of $\theta$. We can differentiate both sides of the equation with respect to $\theta$, to yield its $m^{\text{th}}$-order derivatives. This produces the gradient estimates and the Hessian estimates accordingly, both in recursive forms.

Since [38] already provides a similar derivation in the first-order case, we focus on the second-order. Given Eqn (4), we can further differentiate both sides of the equation by $\theta$. The RHS has three terms from Eqn (4). We rewrite the expression:

**The first term.** This term produces a single term $\nabla_\theta^2 Q(x_t, \pi_\theta(x_t), g)$.

**The second term.** Note a few useful facts: $\nabla_\theta \rho_t^\theta = \rho_t^\theta \nabla_\theta \log \pi_t$, $\nabla_\theta \delta_t = \gamma \nabla_\theta \widehat{V}_{\text{DR}}^{\pi_\theta}(x_{t+1}, g)$. This produces

$$\nabla_\theta \left( \rho_t^\theta \delta_t \nabla_\theta \log \pi_t \right) = \rho_t^\theta \nabla_\theta \log \pi_t \delta_t \nabla_\theta \log \pi_t^T + \gamma \rho_t^\theta \nabla_\theta \log \pi_t \nabla_\theta \widehat{V}_{\text{DR}}^{\pi_\theta}(x_{t+1}, g)^T + \rho_t^\theta \delta_t \nabla_\theta^2 \log \pi_t.$$

**The third term.** Finally, the third term produces

$$\gamma \rho_t^\theta \nabla_\theta^2 \widehat{V}_{\text{DR}}^{\pi_\theta}(x_{t+1}, g) + \gamma \rho_t^\theta \nabla_\theta \widehat{V}_{\text{DR}}^{\pi_\theta}(x_{t+1}, g) \nabla_\theta \log \pi_t^T.$$

Combining all three terms produces the recursive form of the DR Hessian estimates.

$\square$

**Proposition 3.4.** Assume **(A.1)** and **(A.2)** hold. Also assume $\|\pi_\theta - \mu\|_1 \leq \varepsilon = (1 - \gamma)/\gamma$. For any tensor $x$, define $\|x\|_\infty := \max_i |x[i]|$. The $K^{\text{th}}$-order TayPO objective produces the following bias in estimating high-order derivatives,

$$\left\| \mathbb{E}_\mu \left[ \nabla_\theta^m \widehat{V}_K^{\pi_\theta} \right] (x_0) - \nabla_\theta^m V^{\pi_\theta}(x_0) \right\|_\infty \leq \sum_{k=K+1}^\infty \left\| \nabla_\theta^m U_k^{\pi_\theta}(x_0) \right\|_\infty. \tag{9}$$

Hence the upper bound for the bias decreases as $K$ increases. Importantly, when on-policy $\mu = \pi_\theta$, the $K^{\text{th}}$-order TayPO objective preserves up to $K^{\text{th}}$-order derivatives for any $K \geq 0$,

$$\mathbb{E}_\mu \left[ \nabla_\theta^m \widehat{V}_K^{\pi_\theta}(x_0) \right] = \nabla_\theta^m V^{\pi_\theta}(x_0), \forall m \leq K. \tag{10}$$

*Proof.* We can express the residual of the derivatives as

$$\mathbb{E}_\mu \left[ \nabla_\theta^m \widehat{V}_K^{\pi_\theta}(x_0, g) \right] - \nabla_\theta^m V^{\pi_\theta}(x_0, g) = \nabla_\theta^m \left( \widehat{V}_K^{\pi_\theta}(x_0, g) - V^{\pi_\theta}(x_0, g) \right) = \nabla_\theta^m \left( \sum_{k=K+1}^\infty U_k^{\pi_\theta}(x_0, g) \right).$$

Above, the first equality comes from the unbiasedness of the estimates as well as the exchangability between derivatives and expectations, following similar arguments as those in Proposition 3.1. The second equality comes from the Taylor expansion equality in Proposition 3.3. By the assumption that the MDP terminates within an horizon of $H < \infty$, we deduce that the summation contains at most $H - K$ terms and it is valid to exchange derivatives and the summation. Eqn (9) is hence proved by applying the triangle inequality.

$$\left\| \mathbb{E}_\mu \left[ \nabla_\theta^m \widehat{V}_K^{\pi_\theta}(x_0, g) \right] - \nabla_\theta^m V^{\pi_\theta}(x_0, g) \right\|_\infty \leq \sum_{k=K+1}^\infty \left\| \nabla_\theta^m U_k^{\pi_\theta}(x_0, g) \right\|_\infty.$$

When on-policy, we plug in $\pi_\theta = \mu$. Since $K + 1 > m$, this implies that after differentiating $U_k^{\pi_\theta}(x_0, g)$ for a total of $m$ times, each term contains at least $k - m$ terms of $\rho_t^\theta - 1$ for some $t$. Since $K + 1 > m$, this means $U_k^{\pi_\theta}(x_0, g) = 0$ for all indices $k$ within the summation. Hence we have zeros on the RHS and this shows Eqn (10). $\square$

# E   Further details on sampled-based TayPO-$K$ estimates

Please refer to the TayPO [29] paper for further theoretical discussions. By definition, the $K^{\text{th}}$-order increment is

$$U_K^{\pi_\theta}(x_0, g) := \mathbb{E}_\mu \left[ \underbrace{\sum_{t_1=0}^\infty \sum_{t_2=t_1+1}^\infty \cdots \sum_{t_K=t_{K-1}+1}^\infty \gamma^{t_K} \left( \Pi_{i=1}^K (\rho_{t_i}^\theta - 1) \right) Q^\mu(x_{t_K}, a_{t_K}, g)}_{\widehat{U}_K^{\pi_\theta}(x_0, g)} \right].$$

Assume the trajectory is of finite length $T$ (or we can use the effective horizon $T_\gamma = 1/(1 - \gamma)$. The naive Monte-Carlo estimate $\widehat{U}_K^{\pi_\theta}(x_0, g)$ consists of $O(T^K)$ terms. Since we usually care about $K \leq 2$, computing such a term exactly might still be tractable, as is shown later in Appendix F. We show in Algorithm 4 how to compute the estimates with complexity $O(T^2)$.

However, in some applications, we might seek to construct the estimates with better time complexity. The high-level idea is to achieve this through sub-sampling. Define $p_\gamma^\pi(x'|x) := (1 - \gamma) \sum_{t \geq 0} \gamma^t P_\pi(x_t = x'|x_0 = x)$, where $P_\pi$ is the probability measure induced by $\pi$ and the MDP. We can rewrite the above into the following

$$U_K^{\pi_\theta}(x_0, g) = \mathbb{E}_{t_1, t_2 \ldots t_K} \left[ \left( \Pi_{i=1}^K (\rho_{t_i}^\theta - 1) \right) Q^\mu(x_{t_K}, a_{t_K}, g) \right],$$

where the sequence of states are sampled as $x_{t_{i+1}} \sim p(\cdot|x_i', a_i'), a_i' \sim \mu(\cdot|x_i'), x_i' \sim p_\gamma^\mu(\cdot|x_{t_i})$ and $x_{t_0} = x_0$. Note that the above procedure could be achieved by first generating a full trajectory under $\mu$, and then sub-sampling random times along the trajectory. As such, the estimate takes linear time to compute, at the cost of potentially larger variance.

## F  High-level code for implementations of Hessian estimates and meta-gradients

Below, we introduce a few important details on how to convert off-policy evaluation estimates into Hessian estimates, with the help of auto-diff.

### F.1  Implementing Hessian estimates with off-policy evaluation subroutines

In Figure 3, we show a high-level JAX implementation of estimating Hessians using off-policy evaluaition subroutines. The pseudocode assumes access to some well-established off-policy evaluation functions, as are commonly implemented in off-policy RL algorithms such as ACER [5], Retrace [44], IMPALA [47], R2D2 [48] and so on. The function needs to be written in auto-differentiation libraries, such that after the computations, value function estimates' parameter dependencies are naturally maintained. Then this pipeline could be directly implemented as part of a meta-RL algorithm.

```python
# off-policy evaluation subroutine
def off_policy_evaluation(pis, mus, states, actions, rewards, Vs):
    """
    pis: target policy for sampled state-action pairs, shape [T, 1]
    mus: behavior policy for sampled state-action pairs, shape [T, 1]
    states: sampled states, shape [T, S]
    actions: sampled actions, shape [T, A]
    rewards: sampled rewards, shape [T, 1]
    Vs: bootstrapped value functions, shape [T, 1]
    """

    ...
    return evaluation

# jax evaluation function
def evaluation_fun(logits, args):
    """
    compute estimates given target policy defined by the logits
    """
    # softmax parameterization
    pi = jax.nn.softmax(logits, -1)
    # index the policy
    pis = pi[states, actions]
    # evaluation
    evals = off_policy_evaluationn(pis, *args)
    return evals

# gradient and hessian wrt the logits
gradient = grad(evaluation_fun)
hessian = jacfwd(jacrev(evaluation_fun))
```

Figure 3: JAX-based high-level code for the implementation of Hessian estimates. We can easily convert any established trajectory-based off-policy evaluation subroutine into estimates of Hessian matrix, by auto-differentiating through the estimates. This can be implemented in any auto-differentiation frameworks.

### F.2  Examples of off-policy evaluation estimates

The following off-policy evaluation estimates can be abstracted as functions that take as input: a partial trajectory $(x_t, a_t, r_t)_{t=0}^{T}$ of length $T + 1$, the target policy $\pi_\theta$ and a behavior policy $\mu$. Optionally, the function could also take as input some critic function $Q$. We detail how to compute certain estimates below.

**DR estimates.**  In Algorithm 2, we provided the pseudocode for computing DR estimates. The step-wise IS estimates could be computed as a special case by setting $Q = 0$.

**TayPO-**1 **estimates.**  See Algorithm 3 for details.

**TayPO-**2 **estimates.**  See Algorithm 4 for details.

---

**Algorithm 3** Example: an off-policy evaluation subroutine for computing the TayPO-1 estimate

---

**Require: Inputs**: Trajectory $(x_t, a_t, r_t)_{t=0}^T$, target policy $\pi_\theta$, behavior policy $\mu$, (optional) critic $Q$.

Initialize $\widehat{V} = Q(x_T, \pi_\theta(x_T), g)$.

Compute IS ratio $\rho_t^\theta = \pi_\theta(a_t|x_t)/\mu(a_t|x_t)$.

Compute Q-function estimates for all $Q^\mu(x_t, a_t)$. This could be done by computing $\widehat{Q}^\mu(x_t, a_t) = \sum_{s \geq t} r_s \gamma^{s-t} + \gamma^{T-t} Q(x_T, a_T)$.

Compute the estimate $\widehat{V} = \widehat{Q}^\mu(x_0, a_0) + \sum_{t=0}^T \gamma^t (\rho_t^\theta - 1)\widehat{Q}^\mu(x_t, a_t)$.

Output $\widehat{V}$ as an estimate to $V^{\pi_\theta}(x_0, g)$.

---

---

**Algorithm 4** Example: an off-policy evaluation subroutine for computing the TayPO-2 estimate

---

**Require: Inputs**: Trajectory $(x_t, a_t, r_t)_{t=0}^T$, target policy $\pi_\theta$, behavior policy $\mu$, (optional) critic $Q$.

Initialize $\widehat{V} = Q(x_T, \pi_\theta(x_T), g)$.

Compute IS ratio $\rho_t^\theta = \pi_\theta(a_t|x_t)/\mu(a_t|x_t)$.

Compute Q-function estimates for all $Q^\mu(x_t, a_t)$. This could be done by computing $\widehat{Q}^\mu(x_t, a_t) = \sum_{s \geq t} r_s \gamma^{s-t} + \gamma^{T-t} Q(x_T, a_T)$.

Compute the first-order estimate $\widehat{V}_1 = \widehat{Q}^\mu(x_0, a_0) + \sum_{t=0}^T \gamma^t (\rho_t^\theta - 1)\widehat{Q}^\mu(x_t, a_t)$.

Compute the second-order estimate $\widehat{V}_2 = \sum_{t=0}^T \sum_{s=t+1}^T \gamma^s (\rho_t^\theta - 1)(\rho_s^\theta - 1)\widehat{Q}^\mu(x_s, a_s)$.

Output $\widehat{V}_1 + \widehat{V}_2$ as an estimate to $V^{\pi_\theta}(x_0, g)$.

---

**Truncated DR estimates.** See Algorithm 5 for more details. Truncated DR is similar to DR except that the IS ratio is replaced by truncated IS ratios $\min(\rho_t^\theta, \overline{\rho})$ for some $\rho$. For the experiments we set $\rho = 1$, inspired by V-trace [44, 47]. The main motivation for the truncation is to control for the variance induced by IS ratios. However, this also introduces bias into the estimates, unless the samples are near on-policy.

---

**Algorithm 5** Example: an off-policy evaluation subroutine for computing the truncated DR estimate

---

**Require: Inputs**: Trajectory $(x_t, a_t, r_t)_{t=0}^T$, target policy $\pi_\theta$, behavior policy $\mu$, (optional) critic $Q$.

Initialize $\widehat{V} = Q(x_T, \pi_\theta(x_T), g)$.

**for** $t = T - 1, \ldots 0$ **do**

Compute truncated IS ratio $\widetilde{\rho}_t^\theta = \min(\pi_\theta(a_t|x_t)/\mu(a_t|x_t), \rho)$ for some $\rho > 0$.

Recursion: $\widehat{V} \leftarrow Q(x_t, \pi_\theta(a_t), g) + \gamma \widetilde{\rho}_t^\theta (r_t + \gamma Q(x_{t+1}, \pi_\theta(x_{t+1}), g) - Q(x_t, a_t)) + \gamma \widetilde{\rho}_t^\theta \widehat{V}$.

**end for**

Output $\widehat{V}$ as an estimate to $V^{\pi_\theta}(x_0, g)$.

---

### F.3 Implementing meta-RL estimates

To implement meta-RL estimates in a auto-differentiation framework, the aim is to construct a single scalar objective, the auto-differentiation of which produces an estimate to the meta-gradient.

Let $\widehat{V}_{\text{inner}}(\theta, D)$ be an inner loop objective one can use to construct the inner loop adaptation steps. In this case, $\widehat{V}_{\text{inner}}(\theta, D)$ can take as input: the parameter $\theta$ and some data $D$. Here, for example, the data $D$ might consist of sampled trajectories and other hyper-parameters such as discount factors. In our paper, this objective can be replaced by any off-policy evaluation estimates. The updated parameter is computed as: $\theta' = \theta + \eta \nabla_\theta \widehat{V}_{\text{inner}}(\theta, D)$.

The meta objective is defined as the value function, or equivalently some outer loop objective $\widehat{V}_{\text{outer}}(\theta, D)$ that also takes as input the parameter $\theta$ and some data $D$. The overall objective can be defined as:

$$\widehat{V}_{\text{outer}}\left(\theta + \eta \nabla_\theta \widehat{V}_{\text{inner}}(\theta, D)\right).$$

Auto-differentiaing through the above objective can produce estimates to meta-gradients. This objective is also easy to implement in auto-differentiation frameworks.

# G  Experiment

Below, we introduce further details in the experiments.

## G.1  Tabular MDP

**MDPs.**  These MDPs have $|\mathcal{X}| = 10$ states and $|\mathcal{A}| = 5$ actions. The transition probabilities $p(\cdot|x, a)$ are generated from independent Dirichlet distributions with parameter $(\alpha, ...\alpha) \in \mathbb{R}^{|\mathcal{X}|}$. Here, we set $\alpha = 0.001$. The discount factor is $\gamma = 0.8$ for all problems.

**Setups.**  The policy $\pi_\theta$ is parameterized as $\pi_\theta(a|x) = \exp(\theta(x, a))/\sum_b \exp(\theta(x, b))$. The behavior policy $\mu$ is uniform and $\theta$ is set such that $\theta(x, a) = \log \pi(a|x)$ where $\pi = (1 - \varepsilon)\mu + \varepsilon\pi_d$ for some deterministic policy $\pi_d$ and parameter $\varepsilon \in [0, 1]$.

**Experiments.**  In each experiment, we generate a random MDP and initialize the policy. The agent colelcts $N$ trajectories of length $T = 20$ (such that $\gamma^T \approx 0$) from a fixed initial state $x_0$. We then compute gradient and Hessian estimates of the initial state $V^{\pi_\theta}(x_0)$ by directly diffrentiating through various $N$-trajectory off-policy evaluation estimates: $\nabla_\theta^m \widehat{V}^{\pi_\theta}(x_0)$ for $m = 1, 2$. We also calculate the ground truth gradient and Hessian using transition probabilities of the MDPs.

**Accuracy measure.**  Given an estimate $x \in \mathbb{R}^L$ and a ground truth value $y \in \mathbb{R}^L$, we measure the accuracy between the two tensors as:

$$\mathrm{Acc}(x, y) := \frac{x^T y}{\sqrt{x^T x}\sqrt{y^T y}}.$$

Note that this measure is bounded between $[-1, 1]$. This advantage of this measure is that it neglects the absolute scales of the tensors, i.e., if $x = ky$ for some scalar $k \neq 0$, then $\mathrm{Acc}(x, y) = 1$. This metric is used in a number of prior work [21–23] and is potentially a more suitable measure of accuracy given that in large scale experiments, downstream applications typically use adaptive optimizers.

**Further results: effect of sample size.**  In Figure 1(a), we fix the level of off-policyness $\varepsilon = 0.5$ and show the estimates as a function sample size $N$. As $N$ increases, the accuracy measures of most estimates increase. Intuitively, this is because the variance decreases while the bias is not impacted by the sample size. Comparing the step-wise IS estimate with the DR estimate, we see that the DR estimate generally performs better due to variance reduction. This is consistent with findings in prior work [22, 23]). Further, it is worth noting the first-order estimate performs quite well when the $N$ is small, outperforming the DR estimate. This implies the importance of controlling the number of step-wise IS ratios for further variance reduction. However, as $N$ increases, the performance of the first-order estimate does not improve as much compared to other alternatives, and is finally surpassed by the DR estimate mainly due to bigger bias.

Consider the second-order estimate. When $N$ is small, the second-order estimate slightly underperforms the first-order estimate. This is expected because at small sample sizes, the variance dominates. However, as $N$ increases, its performance quickly tops across all different estimates, including the DR estimate. Overall, we expect the second-order estimate to achieve a better bias-variance trade-off, especially in the medium data regime. This should be more significant in large-scale setups where estimation horizons are longer and variance dominates further.

## G.2  Large scale experiments

In large scale experiments, including the continuous 2-D navigation environments and simulated locomotion environments, we adopt the following setups.

**Code base.**  We adopt the code base released by [24]. We make minimal changes to the code base, such that the second-order estimate is comparable to other algorithms under the established experimental setups. For missing hyper-parameters, please refer to the code base for further details. Importantly, note that in the original code base as well as the paper [24], the authors suggest the default learning rate of $\alpha = 10^{-3}$, which we find tends to destabilize learning. Instead, we use $\alpha = 10^{-4}$, which works more stably.

**Computational resources.**  All high-dimensional experiments are run on a computer cluster with multiple CPUs. Each separate experiment is run with 12 CPUs as actors for data collection and parallel computations of parameter updates. The run time for each experiment is on average 36 hours per experiment. For small experiments, we run them on a single laptop machine with 8 CPUs.

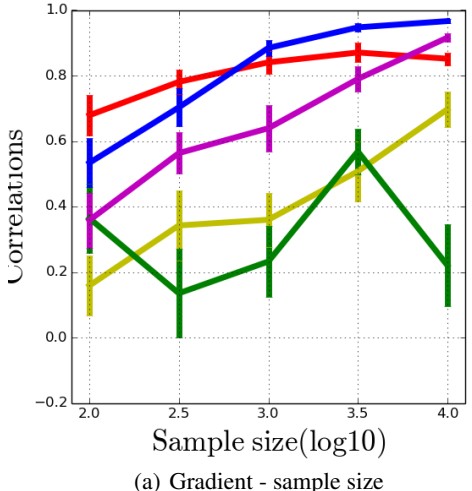

(a) Gradient - sample size

Figure 4: Performance measure as a function of sample size . Each plot shows the accuracy measure between the estimates and the ground truth. Overall, the second-order estimate achieves a better bias and variance trade-off. Here, the plot shows results for estimating gradients.

**Agent details.** The agent adopts a MLP architecture with $[64, 64]$ hidden units. The agent takes in a state $x$ and outputs a Gaussian policy $a \sim \mathcal{N}(\mu_\theta(x), \sigma^2(x))$ where $\mu_\theta, \sigma_\theta$ are parameterized by the neural networks. The agent collects samples $n = 40$ goals at each iteration to construct meta-gradient estimates; the inner loop adaptation is computed with step size $\eta = 0.1$. Inner loop adaptations are computed with $B = 20$ trajectories each of length $T = 100$. All outer loop optimizers use the learning rate $\alpha = 10^{-4}$.

**Algorithmic details.** The PROMP and PROMO-TayPO-2 enforces a soft trust region constraint through clipping

$$\overline{\rho}_t^\theta = \text{clip}(\rho_t^\theta, 1 - \varepsilon, 1 + \varepsilon),$$

where by default $\varepsilon = 0.3$. The PPO optimizers take 5 gradient steps during each iteration. All outer loop gradient based optimizers use Adam optimizers [52].

**Summary of baselines.** The baselines include the following: TRPO-MAML uses TRPO as the outer loop optimizer [45] and the biased MAML implementation [4]; TRPO-FMAML, which is short for first-order MAML, approximates the Hessian matrix by an identity matrix [4]; TRPO-EMAML augments the MAML loss function by an exploration bonus term, which effectively corrects for the bias introduced by vanilla MAML [32, 33]; TRPO-DiCE, which uses the DR estimate to implement the inner loop update, such that the Hessian estimates are unbiased [21–23].

Closely related to our new algorithm is the proximal meta policy search (PROMP) [24], which uses PPO as the outer loop optimizer [46] and the first-order estimate (LVC estimate) as the inner loop loss function [28, 29]. Our new algorithm is called PROMP-TayPO-2, which is a combination of PROMP and TayPO-2. The only difference between PROMP and PROMP-TayPO-2 is that the inner loop loss function is now implemented with the second-order estimate to alleviate the bias introduced by the first-order term.

**Practical implementations of second-order estimates.** We denote $\widehat{V}_1$ as value function estimate based on the first-order approximation; and let $\widehat{V}_2$ be the value function estimate. In practice, the second-order estimate we implement is a mixture between the two estimates with $\beta \in [0, 1]$

$$\widehat{V}_\beta = (1 - \beta)\widehat{V}_1 + \beta\widehat{V}_2.$$

This overall estimate is a convex combination of the two estimates. By moving $\beta = 0$ to $\beta = 1$, we interpolate between the first-order and the second-order estimate. Throughout the large-scale experiments we find $\beta = 0.3$ to work the best. We also show in the next section the sensitivity of performance to $\beta$ on 2-$D$ control environments.

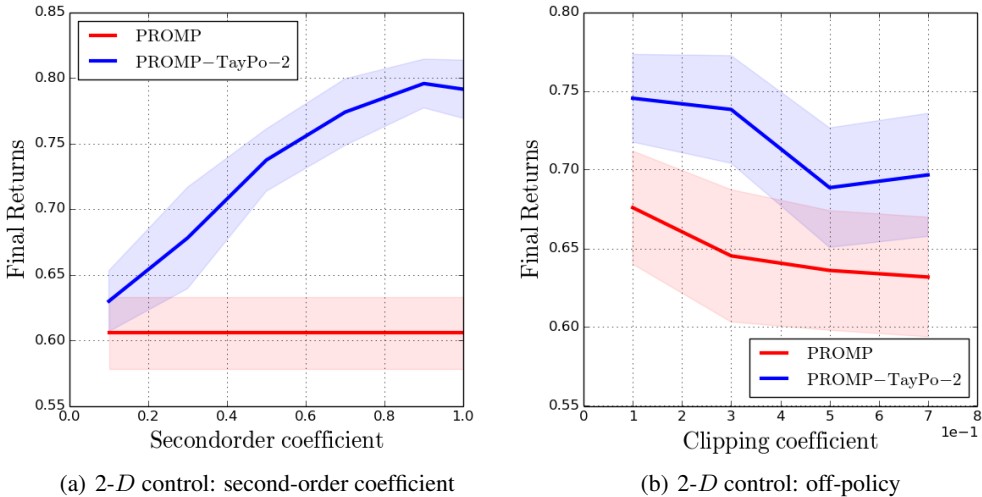

(a) 2-$D$ control: second-order coefficient  (b) 2-$D$ control: off-policy

Figure 5: Ablation study: (a) second-order coefficients; (b) off-policyness. The above two plots show the sensitivity of the first and second-order estimate to hyper-parameters, for 2-$D$ control. The second-order estimate is generally more robust. All curves are averaged over 10 runs.

### G.3 Ablation study

**Sensitivity to the second-order coefficient on 2-$D$ control environments.** In Figure 5(a), we show the sensitivity of the performance to the mixing coefficient $\beta \in [0, 1]$. Note that when $\beta = 0$, the estimate is exactly the first-order estimate; when $\beta = 1$, the estimate recovers the full second-order estimate. We see that on the 2-$D$ control environment, as we increase $\beta$ from 0 to 1, the performance stably improves. When $\beta \approx 0.9$, it seems that the performance reaches a plateau, indicating that $\beta = 0.9$ potentially achieves the best bias and variance trade-off between the two extremes.

**Sensitivity to off-policyness on 2-$D$ control environments.** In Figure 5(b), we study the sensitivity of algorithms to off-policyness in high-d setups. In PROMP, the policies are optimized with behavior policy $\mu$, whose distance against the target policy is constrained by a trust region. The trust region is enforced softly via a clipping coefficient $\varepsilon$ (see Appendix G or [24] for details). When $\varepsilon$ increases, the effective level of off-policyness increases. We see that as $\varepsilon$ increases, both first and second-order estimates degrade in performance. However, the second-order estimates perform more robustly, as similarly suggested in the toy example.