# OpenReview forum: "Unifying Gradient Estimators for Meta-Reinforcement Learning  via Off-Policy Evaluation"
_NeurIPS.cc/2021/Conference — NeurIPS 2021 Poster_

### Official Review · Reviewer_eXfG · 2021-07-13

**Rating:** 6
**Confidence:** 1

**Summary:**

The paper is concerned with meta reinforcement learning. The author(s) introduced a framework for learning high-order derivatives of value functions using off-policy evaluation. Specifically, the author(s) proposed to adopt existing off-policy evaluation methods to derive the value estimator and then take the derivative with respect to the target parameters to construct the estimate. This technique is further adopted to implement meta reinforcement learning.

**Limitations And Societal Impact:**

The authors have discussed the limitations of their work on Page 6.

**Main Review:**

I have a few major comments regarding the proposed procedure.

1. The author(s) proposed to adopt importance sampling and doubly-robust estimators to evaluate the value function. These estimators suffer from the curse of horizon, leading to the sub-optimality of the learned optimal policy. Given that the author(s) considered an MDP setting, there are more efficient estimators for value evaluation (see, e.g., https://arxiv.org/abs/1810.12429; https://arxiv.org/abs/1909.05850).

2. Instead of constructing estimators for the value, it might be more efficient to directly evaluate the gradient of the value and its Hessian matrix. In the single agent setting, there are efficient off-policy policy gradient estimators developed in the literature (see, e.g., https://arxiv.org/abs/2002.04014).

3. I think the author(s) focused on offline meta reinforcement learning problems. It remains unclear to me how the proposed method would work when the positivity assumption (e.g., the policy ratio shall be bounded away from infinity) in the offline dataset is violated. In the single agent setting, the pessimism principle is commonly adopted to mitigate this issue. Would it be useful to combine the pessimism principle with the proposed method?

**Time Spent Reviewing:**

3h

---

> ### Author Response · Authors · 2021-08-10
> **Reply to your reviews**
>
> Many thanks for your time in reviewing our manuscript and your valuable inputs! We reply to specific points in the reviews below.
>
> --- “Importance sampling and doubly robust estimators… suffer from the curse of horizon...”
>
> Thanks for your comments on this. We think there might be a misunderstanding regarding the main proposal of our paper.
>
> We want to stress that we **do not** propose to use importance sampling (IS) or doubly robust (DB) estimator to derive meta gradient estimators. These two estimators are merely shown as special cases which can recover prior work on meta gradient estimates, as summarized in Table 1. In fact, it is exactly because of the curse of horizon & big variance of such methods, that their corresponding meta gradient estimates do not perform well in practice.
>
> It is also exactly because we seek to control the variance of the estimates, that our focus and the new proposal is on biased methods built on Taylor approximation (TayPO) by Tang et al (2020). TayPO explicitly drops the IS ratios in the original IS estimators, such that the overall estimate has small variance. Note that as shown in Tang et al (2020), explicitly dropping IS ratio is also an effective and commonly used method to break the curse of horizon, as implemented in many first-order policy search methods [1,2,3].
>
> [1] Approximately optimal approximate RL, Kakade et al, 2002
>
> [2] Trust region policy optimization, Schulman et al, 2015
>
> [3] Proximal policy optimization, Schulman et al, 2017
>
> The related work you mentioned on bypassing the curse of horizon is based on the idea of marginalized IS. These methods require learning the ratios of stationary distribution, which might introduce more complication in the algorithm (such as solving saddle point optimization problems). Still, we think these are interesting future directions.
>
> In this work, we focus on Taylor approximation as another way to reduce variance, mainly due to its close connections with prior work on meta gradient estimation (note that in **Table 1**, we show TayPO with K=1 recovers an important prior work LVC as a special case).
>
> --- “it might be more efficient to directly evaluate the gradient of the value and its Hessian matrix”
>
> A major motivation of our work is that we can conceptually unify a number of important prior meta gradient estimation methods from the perspective of **directly** differentiating through the value estimators. Our contributions are mainly to build up this connection, and propose to differentiate a new estimator which might lead to higher accuracy in the Hessian estimates.
>
> That being said, we agree that maybe directly constructing estimates to the Hessian can lead to even better results. We would consider this as an interesting future direction for further study. Note that the reference paper you provided only studies policy gradient, and it should be non-trivial to extend the results to Hessian estimation. It is also not clear how to extend such statistically efficient methods to high-dimensional setup, which is also a major motivation for our work. We will provide more concrete discussion in the revision.
>
> --- “I think the authors focused on the offline meta RL problem...”
>
> We want to stress that our focus is **not** on the offline meta RL problem. Throughout the paper, we have focused on the most traditional “online” meta RL setup, where the agent learns by interacting with the environment to collect new data. Note that even in the more “online” setting, where the agent is allowed to interact with the environment, it is still common to assume that the importance sampling ratio is bounded for off-policy learning. See [1,2] listed below. In practice, this can be enforced by learning from behavior policy not too far from the target policy (by e.g., enforcing a trust region constraint).
>
> In practice, when such assumptions are violated, one might need additional techniques to bypass the numerical instability, such as truncation [1]. That being said, there are cases where the practical algorithm works well even with arbitrary behavior policy (i.e., the IS ratio is unbounded), see [2] as an example.
>
> [1] Safe and sample efficient off-policy learning, Munos et al, 2016
>
> [2] AlgaeDice: policy gradient from arbitrary experience, Nachum et al, 2019
>
> As you suggested, the pessimism principle is used in offline RL. Since we are in the “online” setup where the agent interacts with the environment to keep learning, the pessimism principle might be less relevant. However, we agree that this might be valuable in other offline meta RL setups but this is not the focus of our work.

---

### Official Review · Reviewer_DoN5 · 2021-07-14

**Rating:** 7
**Confidence:** 3

**Summary:**

The paper uses the recently developed framework of Taylor expansions for value functions (ref [29] in the paper), expands the results, and shows how they could be applied to Off-Policy Evaluation, and by a non-trivial extension to meta-learning. The main algorthmic contribution of the paper is the application of the ideas to meta-learning, but I feel the main contribution is a conceptual one, further developing the ideas from [29] and drawing theoretical connections between OPE and meta-learning.

The paper includes an experimental section demonstrating marginal improvement over baselines in several domains.

**Limitations And Societal Impact:**

The authors do a good job of discussing the limitations of their method compared to alternative methods.

The paper is mostly theoretical and I don't see a potential negative soceital impact.

**Main Review:**

Overview: Despite some flaws, I think this is a good paper and leaning towards acceptance. I think the two main limitations of the paper are:
(1) It draws very heavily on the theory and novelty of [29] and therefore its own novelty is a bit limited.
(2) The empirical improvements obtained for meta-learning (the problem the paper sets out to advance) are limited.
Additionally, there are some changes that should be made to make the paper clearer, but I think those can be easily fixed for the camera ready if the paper is accepted. Despite the drawbacks, the reason I support acceptance is that I feel the additional contributions on top of the already interesting results of [29] are non-trivial, and this is a potentially interesting research direction which could provide insight into many aspects of RL. For example, viewing meta-learning through the lens of Taylor expansions for value functions allows for a very clear connection and unification of derivatives estimations of value functions and OPE.

A general suggestion for the authors which I feel might improve the overall story of the paper, is to lean more heavily on the theoretical results and their ability to shed light on interesting insights such as the connection between OPE and meta-learning, and OPE to high order derivatives. At least my personal experience reading the paper was expecting significant novelty and improvement of results on meta-learning, which set me up for a disappointment. It was only on a second read that I appreciated the theoretical insights in the first half of the paper more. Additionally, I would set up the connection with [29] more - the authors do a good enough job of properly citing [29], but not enough to make it clear that the reader should be familiar with [29] to fully appreciate this paper. After emphasizing the theory and insights, I think the meta-learning algorithm and result can be presented as a cool implication (this is obviously a very subjective suggestion).

Some more specific comments:
- In lines 120-122 regarding how the approximation maintains the higher-order dependencies on \theta. I feel like I'm missing something and the sentence is failing to convey an idea it is trying to. Obviously when you make an approximation in some parameter you maintain higher order terms in the small parameter, and choose how many terms to keep. So this statement seems trivial and I'd like to know if there's something deeper which the authors are trying to convey and I'm missing.
- In section 3.2 and Proposition 3.1 - Is K a typo and should be m? (if yes, it is a consistent typo, which is why I'm confused). If it isn't a typo, I couldn't see where K is defined, and also can't follow the proof of Proportion 3.1, so this is a point I would really like clarification on.
- In section 5.1, the authors claim to investigate the bias/variance tradeoff of their method, but only show results for the overall MSE of estimating the derivative. I think the authors should show explicit plots of the decomposed variance and bias of each estimator shown.
- very minor - the y-labels in Figure 2 a-b are cut.

**Time Spent Reviewing:**

4 to 6

---

> ### Author Response · Authors · 2021-08-10
> **Reply to your reviews**
>
> Many thanks for your time in reviewing our manuscript and your valuable inputs! We reply to specific points in the reviews below.
>
> --- “A general suggestion for the authors...”
>
> Thank you very much for your feedback! We appreciate your specific comment on how to improve the story of the paper. Indeed, we think that a major contribution of this work is to build the conceptual connections between OPE and methods for estimating high-order derivatives of value functions.
>
> Here, we intended to draw a direct connection between OPE and meta gradient estimation, instead of implying the connection as a form of downstream application. The main motivation stems from the observation that many important meta gradient estimation methods are directly linked to OPE estimators, especially important is the LVC estimator, which relates to TayPO-1 and naturally allows us to present TayPO-2 as a natural extension.
>
> We are very curious to hear what you think about this and we will certainly make the focus of the paper more clear in the revision.
>
> --- “line 120 - line 122”
>
> Thank you for pointing out some potential confusions here.
>
> Our intuition is probably better explained with an 1-d example. Think about theta as a scalar, drawn on the x-axis and V(theta) drawn on the y-axis. If we have an OPE estimator \hat{V}(theta) which approximates V(theta) at each value of theta, then intuitively the derivative of \hat{V}(theta) should convey information about the derivative of V(theta) (which is the gradient). By the explanation in line 120-122, we intended to convey the intuition that directly differentiating through the OPE estimator should give us an estimate of high-order derivatives of value functions. We will make this intuition more concrete so that the interpretation is less seemingly trivial.
>
> --- “Is K a typo or should it be m?”
>
> It is a typo, here K should be m and indicates the m-order derivatives. The notation of K is meant to come later when discussing the order of Taylor approximations. We will make the correction.
>
> --- “bias/variance tradeoff of their method”
>
> Thank you for pointing this out. In Sec 5.1, we presented the accuracy measure as the correlation coefficient between the ground truth and the estimator. This is a commonly used metric in a number of prior work (see [21-23] in the paper). We speculate that this might be a more meaningful accuracy measure which is more indicative of the practical impact of the estimates. This is because in practice, adaptive optimizers such as Adam scale the magnitude of the gradient estimates, making their angular distance against the ground truth gradient more important.
>
> Still, we are very happy to provide detailed plots of the bias/variance study of various estimators in the revision.
>
> -- other typos with the text and figures
>
> We will correct them in the revision.

---

### Official Review · Reviewer_WYwM · 2021-07-15

**Rating:** 4
**Confidence:** 5

**Summary:**

This paper uses Huang and Jiang (2020) to talk about gradient-based meta learning in reinforcement learning using the language of off-policy evaluation. The logical connection behind those concepts stands from the fact that policy gradient methods in reinforcement learning are in fact instances of the likelihood ratio derivative estimator, which itself stems from important sampling. Proposition 3.2 and 3.3 are obtained from Huang and Jiang. Proposition 3.1 tries to extend their results to talk about higher order derivatives. The author's finally propose to use ideas from Taylor policy optimization from Tang et al. (2020) as a variance reduction method within the doubly robust policy gradient framework of Huang and Jiang.




**Limitations And Societal Impact:**

The limitation of the framework per se are not quite explored. The authors talk about what can be said within that framework (dice) but not quite what it cannot say. Experimentally the paper looks at the performance of the combination of doubly robust plus Taylor approximation as an algorithm of its own. It does not quite try to understand what the doubly robust formulation allows us to address specifically in meta learning. It would be interesting to gain more insights into the value of the proposed method by taking a more ablative approach. Since the authors pitch a framework as the main contribution, performance results matter much less than experimental insights at the conceptual level.  I think that the paper could be restructured around the algorithm and to de-emphasize the framework part. To go all in on the algorithm (which might be how things happened chronologically anyways).

**Main Review:**

In my opinion the paper doesn't go deep enough into the individual ideas being put forward. The doubly robust estimator of Huang and Jiang is at the core of this paper.  In the specific context of meta learning, what does it allow us to say or do that we couldn't achieve before? Why is doubly robust necessary? The doubly robust being a variance reduction technique why this VRT in particular and not another one, in the meta learning context? Regarding the Taylor approximation, why is it related to the doubly robust estimator? The paper does not do a convincing job at showing that these two components strongly depend on each other: conceptually, or experimentally. Overall, the paper lacks cohesion and depth.

On line 83 the notation suggests that the expectation is not taken over x and only over the task g. While technically correct, you don't really have to write the objective as an expectation over the value function. Your objective should simply be the expected discounted or on discounted return, written plainly as a sum of rewards.

In equation 2, I would suggest using a different notation when you're writing nabla of theta prime. Theta prime should be a function, but the way that it's written suggests that it's a variable instead.

Another suggestion for notation would be to write "v of theta" (instead of a superscript) in equation 2. This way it would be clearer how you have to apply the chain rule later on.

Reading more carefully on line 92 and equation 1 now I realize that you assume a fixed initial distribution starting from x0. This is confusing because on line 83 you simply write v of x and it looks like x should be a random variable: your initial state. Perhaps you should state explicitly that you assume (without loss of generality) a given initial state.

On line 92, the expression the "Hessian of the value function" is misleading because really the Hessian that you're taking is that of the expected return objective, not that of the value function.

I think that the unnumbered equation below line 106 is redundant: it simply says that the two quantities are unbiased estimators. Words suffice here. Instead, I would suggest that you explain the nature of the issue. The correlation problem that you're describing comes from taking the expectation over a product of two random variables. The fact that it's not simply the product of the individual expectations and that here is also a covariance term.

I'm confused by proposition 3.1 especially around line 137 when you introduce a new notation for the estimator nabla hat v. What you have in this proposition should be something about a derivative estimator: score function/REINFORCE, reparametrization. Now you don't seem to be defining how you obtain this nabla theta m  v hat, and in that sense proposition 3.1 does not quite stand on its own.

On line 171 you allude to the fact that the variance of policy learning is essentially due to having a large number of importance sampling terms. While this is a reasonable intuition to have, note that we can show formally that PDIS-type estimators need not have larger variance than let's say stationary or marginal IS.


**Time Spent Reviewing:**

1

---

> ### Author Response · Authors · 2021-08-10
> **Reply to your reviews**
>
> Many thanks for your time in reviewing our manuscript and your valuable inputs! We reply to specific points in the reviews below.
>
> --- **General response**
>
> We think there may be a misunderstanding from the reviewer regarding the main message and important components of our paper, and so we restate the main contributions below.
>
> The central idea of our paper is to unify seemingly unrelated prior methods on meta gradient estimation, through the lens of off-policy evaluation. To our knowledge, this interpretation is novel and allows for a better understanding of a number of important prior methods (see **Table 1**). This interpretation also allows us to naturally propose a new meta gradient estimator based on TayPO-2.
>
> The doubly robust (DB) estimator introduced to the MDP setting by Jiang & Li and Thomas & Brunskill, and discussed in Huang and Jiang (2019) is an important special case of our framework, since it recovers prior work such as loaded DiCE.  However, it is just one of many methods for off-policy estimation that falls within the framework of this paper. As we explicitly discussed (see e.g., line 165 - line 172), the DB estimator has limitations within this framework due to its unbiased nature. By taking a more general framework that allows for bias-variance trade-offs we can interpret higher-performance meta RL methods such as LVC within our framework too, via special cases of TayPO -- a class of biased estimators.
>
> -- “Prop 3.2 and 3.3 are obtained from Huang and Jiang...”
>
> This is not true. Though Proposition 3.2 is a generalization of the result from (Huang and Jiang, 2019), Proposition 3.3 is not. In fact, as explicitly stated in the paper, the latter proposition is adapted from results from Tang et al., (2020).
>
> -- “Tang et al (2020) as a variance reduction method within the doubly robust policy gradient framework of Huang and Jiang”
>
> This is not true. TayPO is not a ‘variance reduction technique’, it is a stand-alone framework to construct low-variance but biased off-policy evaluation estimators. TayPO in Tang et al (2020) is developed outside of the DB framework, since DB is unbiased and TayPO is by design biased. Lastly, we also do not combine TayPO with DB -- TayPO is always used independently from DB, throughout our derivation and experiments.
>
> --- “The doubly robust estimator is at the core of the paper, what does it allow us to say or do that we couldn’t achieve before? Why is doubly robust necessary?”
>
> As discussed earlier in the general reply to your comments, there is a misunderstanding here.
>
> Firstly, the core of the paper is **not** about the doubly robust (DB) estimator. The core of the paper is the observation that prior work on meta gradient estimation can be derived via different instantiations of off-policy evaluation estimators, as summarized in Table 1. This interpretation allows us to conceptually unify a few seemingly disparate methods in meta gradient estimation, and propose a new variant based on the recently developed TayPo-2 off-policy evaluation estimator.
>
> Again notice, that DB is not the focus of our work. DB is an important special case because it is unbiased, offers variance reduction, and corresponds to a few prior work on meta gradients (e.g., loaded DiCE). However, our focus (and the new algorithm) is based on TayPO (Tang et al, 2020), which are a family of biased off-policy evaluation estimators.
>
> --- “The doubly robust being a variance reduction technique why this VRT and not another one, in the meta learning context?”
>
> We do not explicitly choose variance reduction techniques in our paper. Note that our main contribution is to make the conceptual connections between existing off-policy evaluation estimators and prior meta gradient estimators, summarized in Table 1.
>
> The central insight of DB is to use baseline function for variance reduction. This is arguably the most commonly-used variance reduction technique in general RL literature. We speculate this is the main reason why prior meta gradient literature implicitly uses DB as a variance reduction technique.
>
> --- “Why is the Taylor approximation related to the doubly robust estimator?”
>
> Thank you for pointing out a potential confusion here. We will make this more clear in the revision. Both Taylor approximation (which we will call interchangeably as TayPO below) and DB are ways to construct off-policy evaluation estimators to the value functions. Their main difference lies in that DB is unbiased, while offering variance reduction through the baseline functions. TayPOs are in general biased estimators, while offering an even larger extent of variance reduction by limiting the number of IS ratios. Here, TayPOs are of particular interest mainly because an important prior work (LVC) can be seen as a special case of TayPO when K=1. This naturally leads us to consider other estimators stemming from the TayPO framework.
>
> --- “...not a convincing job at showing that these two components strongly depend on each other: conceptually, or experimentally”
>
> As pointed out before, DB and Taylor approximations are conceptually related because they are both off-policy evaluation estimators, and they are both related to prior work in meta gradient estimation.
>
> Experimentally, we have made explicit comparison between DB and TayPO K=1,2 in Section 5.1. We see that though DB achieves variance reduction due to the baseline (compared to the naive IS estimate), it still suffers more variance compared to TayPO K=1,2. Their impact on practical performance is also made in the experiment section.
>
> --- “theta-prime should be a function”
>
> Thank you for pointing out a potential confusion here. We will make the notation more clear.
>
> --- “fixed initial distribution starting from x0”
>
> We will make sure to clarify this point in the revision. We always assume that the problem has a fixed state x0, though for convenience we can still define value functions and estimates thereof over other states x.
>
> --- “Hessian of value function is misleading because you are taking Hessian of the expected return objective, not that of the value function”
>
> There might be a potential confusion here. In the meta RL framework shown in Eqn (1), the adaptation U(theta,g) is always computed per task. As a result, when deriving the meta gradient formula in Eqn (2), we need to calculate the Hessian of the **value function** per task.
>
> --- “unnumbered equation below line 106 is redundant”
>
> Thanks for your suggestion. We will consider removing this equation in the revision.
>
> --- “confused by proposition 3.1”
>
> Proposition 3.1 is meant to show a general result on generic **unbiased** off-policy evaluation estimator. Here, the estimator \hat{V} is a generic unbiased estimator, and it can be a full IS or DB estimator as special cases. The main message of Proposition 3.1 is that as long as one can construct an unbiased estimate of the value function, and if \hat{V} is m-time differentiable with respect to theta (Assumption 1), then one can construct unbiased estimate to the m-order derivative by **directly** differentiating the estimate \hat{V}.
>
> We do not specify whether this gradient estimate is REINFORCE/score-function or reparameterized, nor is there any need in doing so. This is because the aforementioned recipe of directly differentiating \hat{V} would work, as long as A.1 and A.2 are satisfied, regardless of how the estimates are specifically built. In the special example of DB, the resulting estimate would be score-function based; it would also be possible to construct value function estimates, such that their direct differentiations produce reparameterization based estimates.
>
> --- “We can show formally that PDIS-type estimators need not have larger variance than stationary or marginalized IS”
>
> While PDIS-type estimates do not dominate e.g. marginalized IS [1] in variance across all possible MDP instances, it is arguably true that in many practical setups one would still avoid having many IS ratios in the estimator. This is the main motivation for prior work such as Retrace [2] (where IS ratios are truncated) and TayPO [3] (where IS ratios are dropped). We will certainly make this point more clear.
>
> [1] Breaking the curse of horizon: infinite horizon off-policy estimation, Liu et al 2019
>
> [2] Safe and sample efficient off-policy learning, Munos et al, 2016
>
> [3] Taylor expansion policy optimization, Tang et al, 2019
>
> --- “the authors talk about what can be said within that framework but not quite what it cannot say”
>
> Our framework has reinterpreted a number of important prior work as specific instantiations of differentiating through off-policy evaluation estimates. Yet, we explicitly discuss the limitation in line 214 - line 221, that there are still many meta gradient estimators which work well in practice, but cannot be understood as special cases in our current framework.
>
> --- “experimentally the paper looks at the performance of combination of doubly robust plus Taylor approximation as an algorithm”
>
> Once again, we stress that this is **not** what our algorithm proposes to do. We do **not** combine DB with Taylor approximation in the experiment. Throughout our paper, DB and TayPO are two stand-alone methods for constructing off-policy estimates to value functions. Our new algorithm is based on TayPO-2 (Taylor approximation with K=2), which explicitly maintains up to K=2 IS ratios, and is by design completely distinct from DB which maintains all IS ratios.
>
> In Sec 5, we always separate DB, TayPO-1 and TayPO-2. We compare their performance in both tabular MDP setups and high-dimensional problems.

---

### Official Review · Reviewer_RG8H · 2021-07-16

**Rating:** 6
**Confidence:** 3

**Summary:**

This paper considers meta-RL with gradient-based adaptations, which relies heavily on the estimation of the Hessians of value functions. Though there are previous approaches that focus on unbiased/biased estimations of Hessian. This paper gives a unified view for estimating higher-order derivatives of value functions, through the lens of off-policy evaluation. This subsumes several previous works, also sheds some light on the bias-variance tradeoff of the estimate. Empirically, the new proposed second-order estimate is also incorporated in meta-RL algorithms to showcase the empirical efficiency.

**Ethics Review Area:**

["I don’t know"]

**Limitations And Societal Impact:**

Limitations are discussed in Section 3, while it seems it does not have any societal impact.

**Main Review:**

(Originality): The main contributions I see from this paper are: (1). a unified framework for estimating higher-order derivatives of value functions, through OPE methods. (2). the use of Taylor expansion in the value function, which controls the number of IS ratios in the value estimate, hence sheds some light on the bias-variance trade-off. (3). also a new method arising from this framework.

(Quality): The motivation is clear, and the method is clearly stated. Some comments here include:
[1]. For the bias-variance trade-off with the Taylor expansion, how to select the optimal $K$? Also, how does this bias-variance trade-off translate into the downstream meta-gradient estimations? Any comments on this?
[2]. In experiment section 5.1, the bias-variance trade-off examination seems a little bit confusing to me. It seems the argument here is the second-order method achieves a better trade-off through high correlation with the ground truth. Is there any way to empirically examine the bias and variance in this tabular setting?
[3]. For the continuous control problem, it seems the improvement from the new proposed approach is pretty marginal since the bias might be a dominating issue here. Are there any environments/MDPs/settings that we can expect a larger improvement of it? It would be great to include some discussions/new experiments on this.

(Clarity): The paper is well-written and easy to follow.


**Time Spent Reviewing:**

2 hours

---

> ### Author Response · Authors · 2021-08-10
> **Reply to your reviews**
>
> Many thanks for your time in reviewing our manuscript and your valuable inputs! We reply to specific points in the reviews below.
>
> --- “For the bias variance trade-off with Taylor expansion, how to select the optimal K?”
>
> Thank you for pointing this out. In the paper, we focus on the case where K=2. There are a few reasons: (1) the simplest case K=1 recovers a prior work as a special case, hence K=2 is a natural next step to consider; (2) Tang et al, 2020 shows that K=2 seems to achieve a good balance between bias and variance trade-off in deep RL setup, and is more feasible to implement than K>=3; (3) In the tabular experiments, we also show that K=2 can generally achieve superior performance compared to K=1. Overall, we think K=2 is a natural choice to consider while K>=3 is also possible. In practice, the optimal K might depend on specific environments, but our experience is that K=2 should be the first to consider, and is often sufficient to achieve improvements in performance.
>
> --- “how does this bias-variance trade-off translate into downstream meta gradient estimation?”
>
> The meta gradient estimate is the product between Hessian estimate and a PG estimate (line 95 - line 103). As discussed in line 104 - line 110, since these two estimates are in general correlated, it is difficult to characterize precisely how the Hessian bias-variance translates into downstream meta gradient estimates. However, when sample sizes are large and it is feasible to assume that both estimates can be considered independently of each other, we should expect a good bias-variance trade-off of the Hessian estimate to translate into good downstream estimation. In practice, we should expect such effects even when sample sizes are relatively small. This is corroborated by results in Section 5, where Hessian estimation with better trade-off leads to better downstream performance, implying potentially better meta gradient estimates.
>
> --- “is there any way to empirically examine the bias and variance in this tabular setting?”
>
> Thank you for pointing this out. Indeed, in Section 5.1 our accuracy measure is the correlation between the estimator and the ground truth. We used this measure mainly because this is a major accuracy measure in major prior work (please see [21-23] in the paper). We speculate the reason why correlation measure is more commonly used in this space of literature is as follows: since the correlation measures the angular distance, it might be a more sensible measure because in practice gradient magnitudes are scaled by optimizers such as Adam. This implies that the correlation measure might be more indicative of practical improvement.
>
> Still, we are happy to provide the exact bias-variance measure in the MDP example in the revision.
>
> --- ‘the improvement from the new proposed approach is pretty marginal“
>
> We agree that the improvements on continuous control problems are relatively marginal. We speculate that in this case, the main problem is the variance, so that TayPO K=1 performs well enough (in the sense that TayPO K=1 has large bias but achieves stronger variance reduction than TayPO K>=2). That being said, the performance gains should depend on the hyper-parameter setup of the base algorithm. Throughout the experiments, we use the default hyper-parameter setup of (Rothfuss et al, 2018). It should be the case that under their hyper-parameter setup (e.g., batch size), TayPO K=1 already achieves a good trade-off and TayPO K=2 only provides marginal gains.
>
> We are happy to provide further ablation study on the effect of such hyper-parameters in the revision. We expect when the batch size increases, and when the bias takes up the main effect, TayPO K=2 should achieve more significant gains. This can be corroborated by results in Fig 1(d), where we use the same batch size for the 2-d control task as with other more high-d tasks in Sec 5.3. In this sense, the batch size is relatively large and indeed we observe a bigger performance gap between TayPO-2 and other baselines.

---

### Author Response · Authors · 2021-08-25
**Message to all reviewers**

Hi all reviewers,

Thank you once again for your efforts in providing feedback to our paper.

We would like to ask whether your questions & concerns have been fully addressed, or if there are additional questions that we may answer. We appreciate your further feedback.

Many thanks for your time.

Sincerely,

Authors

---

### Decision · Program_Chairs · 2021-09-27

**Decision:**

Accept (Poster)

**Comment:**

The expert reviewers for the most part appreciated the paper and were mostly positive. One issue that arose in discussion is that the intent and contribution of the paper is not sufficiently clearly stated, leading to some potential misunderstandings; the authors should very carefully use the feedback of all reviewers, both positive and negative, to more clearly frame the idea of bridging of meta-learning gradients and OPE. An addition, given the significant limitations of sequential IS (including as part of DR), the authors should offer a clear discussion of more efficient methods for OPE and gradient estimation such as pointed out by the reviewers and what might the implications of these potentially be in the authors' setting; this is an important discussion even if fully studying these implications in depth may be beyond the scope of the paper. The authors should similarly address the other points raised in the reviews and implement fixes suggested in their response.